# Learning Conjoint Attentions for Graph Neural Nets

**Tiantian He[1,2]  Yew-Soon Ong[1,2]  Lu Bai[1,2]**
[1]Agency for Science, Technology and Research (A*STAR)
[2]DSAIR, Nanyang Technological University
{He_Tiantian,Bai_Lu}@ihpc.a-star.edu.sg, Ong_Yew_Soon@hq.a-star.edu.sg
{tiantian.he,bailu,asysong}@ntu.edu.sg

## Abstract

In this paper, we present Conjoint Attentions (CAs), a class of novel learning-to-attend strategies for graph neural networks (GNNs). Besides considering the layer-wise node features propagated within the GNN, CAs can additionally incorporate various structural interventions, such as node cluster embedding, and higher-order structural correlations that can be learned outside of GNN, when computing attention scores. The node features that are regarded as significant by the conjoint criteria are therefore more likely to be propagated in the GNN. Given the novel Conjoint Attention strategies, we then propose Graph conjoint attention networks (CATs) that can learn representations embedded with significant latent features deemed by the Conjoint Attentions. Besides, we theoretically validate the discriminative capacity of CATs. CATs utilizing the proposed Conjoint Attention strategies have been extensively tested in well-established benchmarking datasets and comprehensively compared with state-of-the-art baselines. The obtained notable performance demonstrates the effectiveness of the proposed Conjoint Attentions.

## 1 Introduction

Graph neural networks (GNNs) have shown much success in the learning of graph structured data. Amongst these noteworthy GNNs, attention-based GNNs [33] have drawn increasing interest lately, and have been applied to solve a plethora of real-world problems competently, including node classification [17, 33], image segmentation [35], and social recommendations [30].

Empirical attention mechanisms adopted by GNNs aim to leverage the node features (node embeddings) to compute the normalized correlations between pairs of nodes that are observed to connect. Treating normalized correlations (attention scores/coefficients) as the relative weights between node pairs, attention-based GNN typically performs a weighted sum of node features which are subsequently propagated to higher layers. Compared with other GNNs, especially those that aggregate node features with predefined strategies [1, 17, 19], attention-based GNNs provide a dynamical way for feature aggregation, which enables highly correlated features from neighboring nodes to be propagated in the multi-layer neural architecture. Representations that embed with multi-layer correlated features are consequently learned by attention-based GNNs, and can be used for various downstream tasks.

Though effective, present empirical graph attention has several shortcomings when aggregating node features. First, the computation of attention coefficients is limited solely to the correlations of internal factors, i.e., layer-wise node features within the neural nets. External factors such as cluster structure and higher-order structural similarities, which comprise heterogeneous node-node relevance have remained underexplored to be positively incorporated into the computation of more purposeful attention scores. Second, the empirical attention heavily leaning on the node features may cause

35th Conference on Neural Information Processing Systems (NeurIPS 2021).

over-fitting in the training stage of neural nets [34]. The predictive power of attention-based GNNs is consequently limited.

To overcome the mentioned challenges, in this paper, we propose a class of generic graph attention mechanisms, dubbed here as Conjoint Attentions (CAs). Given CAs, we construct Graph conjoint attention networks (CATs) for different downstream analytical tasks. Different from previous graph attentions, CAs are able to flexibly compute the attention coefficients by not solely relying on layer-wise node embeddings, but also allowing the incorporation of purposeful interventions brought by factors external to the neural net, e.g., node cluster embeddings. With this, CATs are able to learn representations from features that are found as significant by diverse criteria, thus increasing the corresponding predictive power. The main contributions of the paper are summarized as follows.

- We propose Conjoint Attentions (CAs) for GNNs. Different from popular graph attentions that rely solely on node features, CAs are able to incorporate heterogeneous learnable factors that can be internal and/or external to the neural net to compute purposeful and more appropriate attention coefficients. The learning capability and hence performance of CA-based GNNs is thereby enhanced with the proposed novel attention mechanisms.

- For the first time, we theoretically analyze the expressive power of graph attention layers considering heterogeneous factors for node feature aggregation, and the discriminant capacity of such attention layers, i.e., CA layers is validated.

- Given CA layers, we build and demonstrate the potential of Graph conjoint attention networks (CATs) for various learning tasks. The proposed CATs are comprehensively investigated on established and extensive benchmarking datasets with comparison studies to a number of state-of-the-art baselines. The notable results obtained are presented to verify and validate the effectiveness of the newly proposed attention mechanisms.

## 2   Related works

To effectively learn low-dimensional representations in graph structured data, many GNNs have been proposed to date. According to the ways through which GNNs define the layer-wise operators for feature aggregation, GNNs can generally be categorized as spectral or spatial [38].

**Spectral GNNs**-The layer-wise function for feature aggregation in spectral GNNs is defined according to the spectral representation of the graph. For example, Spectral CNN [3] constructs the convolution layer based on the eigen-decomposition of graph Laplacian in the Fourier domain. However, such layer is computationally demanding. To reduce such computational burden, several approaches adopting the convolution operators which are based on simplified or approximate spectral graph theory have been proposed. First, parameterized filters with smooth coefficients are introduced for Spectral CNN to incorporate spatially localized nodes in the graph [14]. Chebyshev expansion [7] is then introduced to approximate graph Laplacian rather than directly performing eigen-decomposition. Finally, the graph convolution filter is further simplified by only considering first or higher order of connected neighbors [17, 37], so as to make the convolution layer more computationally efficient.

**Spatial GNNs**-In contrast, spatial GNNs define the convolution operators for feature aggregation by directly making use of local structural properties of the central node. The essence of spatial GNNs consequently lies in designing an appropriate function for aggregating the effect brought by the features of candidate neighbors selected based on appropriate sampling strategy. To achieve this, it sometimes requires to learn a weight matrix that accords to the node degree [8], utilize the power of transition matrix to preserve neighbor importance [1, 4, 18, 19, 42], extract the normalized neighbors [24], or sample a fixed number of neighbors [13, 44].

As representative spatial GNNs, attention-based GNNs (GATs) [12, 33] have shown promising performances on various learning tasks. What makes them effective in graph learning is a result of adopting the attention mechanism, which has been successfully used in machine reading and translation [5, 22], and video processing [40], to compute the node-feature-based attention scores between a central node and its one-hop neighbors (including the central node itself). Then, attention-based GNNs use the attention scores to obtain a weighted aggregation of node features which are subsequently propagated to the next layer. As a result, those neighbors possessing similar features may then induce greater impact on the center node, and meaningful representations can be inferred by GATs. Having investigated previous efforts to graph neural networks, we observe that the computation

of empirical graph attentions heavily relies on layer-wise node features, while other factors, e.g., structural properties that can be learned outside of neural net, have otherwise been overlooked. This motivates us in proposing novel attention mechanisms in this paper to alleviate the shortcomings of existing attention-based graph neural networks.

## 3 Graph conjoint attention networks

In this section, we elaborate the proposed Conjoint Attention mechanisms, which are the cornerstones for building layers of novel attention-based graph neural networks. Mathematical preliminaries and notations used in the paper are firstly illustrated. Then, how to construct neural layers utilizing various Conjoint Attention mechanisms are introduced. Given the formulated Conjoint Attention layers, we finally construct the Graph conjoint attention networks (CATs).

### 3.1 Notations and preliminaries

Throughout this paper, we assume a graph $G = \{V, E\}$ containing $N$ nodes, $|E|$ edges, and $C$ classes ($C \ll N$) to which the nodes belong, where $V$ and $E$ respectively represent the node and edge set. We use $\mathbf{A} \in \{0,1\}^{N \times N}$ and $\mathbf{X} \in \mathbb{R}^{N \times D}$ to represent graph adjacency matrix and input node feature matrix, respectively. $\mathcal{N}_i$ denotes the union of node $i$ and its one-hop neighbors. $\mathbf{W}^l$ and $\{\mathbf{h}_i^l\}_{i=1,\dots N}$ denote the weight matrix and features (embeddings) of node $i$ at $l$th layer of CATs, respectively, and $\mathbf{h}^0$ is set to be the input feature, i.e., $\mathbf{X}$. For the nodes in $\mathcal{N}_i$, their possible feature vectors form a multiset $M_i = (S_i, \mu_i)$, where $S_i = \{s_1, \dots s_n\}$ is the ground set of $M_i$ which contains the distinct elements existing in $M_i$, and $\mu_i : S_i \to \mathbb{N}^\star$ is the multiplicity function indicating the frequency of occurrence of each distinct $s$ in $M_i$.

### 3.2 Structural interventions for Conjoint Attentions

As aforementioned, the proposed Conjoint Attentions are able to make use of factors that are either internal or external to the neural net to compute new attention coefficients. Internal factors refer the layer-wise node embeddings in the GNN. While, the external factors include various parameters that can be learned outside of the graph neural net and can potentially be used to compute the attention scores. Taking the cue from cognitive science, where contextual interventions have been identified as effective external factors that may improve the attention and cognitive abilities [16], here we refer these external structural properties as *structural interventions* henceforth for the computing of attention coefficients. Next, we propose a simple but effective way for CAs to capture diverse structural interventions external to the GNNs. Let $\mathbf{C}_{ij}$ be some structural intervention between $i$th and $j$th node in the graph. It can be obtained with the following generic generating function:

$$\mathbf{C}_{ij} = \underset{\phi(\mathbf{C})_{ij}}{\arg\min} \Psi(\phi(\mathbf{C})_{ij}, \mathbf{Y}_{ij}), \tag{1}$$

where $\Psi(\cdot)$ represents a distance function and $\phi(\cdot)$ stands for an operator transforming $\mathbf{C}$ to the same dimensionality of $\mathbf{Y}$. Given the generic generating function in Eq. (1), it is known that many effective paradigms for learning latent features can be used for the subsequent computation of Conjoint Attentions, if the prior feature matrix $\mathbf{Y}$ is appropriately provided. Taking $\mathbf{A}$ as the prior feature matrix, in this paper, we consider two generation processes that can capture two unique forms of structural interventions. Let $\Psi(\cdot)$ be the euclidean distance, when $\phi(\mathbf{C})_{ij} \doteq \mathbf{V}\mathbf{V}_{ij}^T$, we have:

$$\mathbf{C}_{ij} = \underset{\mathbf{V}\mathbf{V}_{ij}^T}{\arg\min} (\mathbf{A}_{ij} - \mathbf{V}\mathbf{V}_{ij}^T)^2, \tag{2}$$

where we use an $N$-by-$C$ matrix $\mathbf{V}$ to approximate $\mathbf{C}$ to reduce the computational burden. As it shows in Eq. (2), $\mathbf{C}_{ij}$ attempts to acquire the structural correlation pertaining to node cluster embeddings, based on matrix factorization (MF). A higher $\mathbf{C}_{ij}$ learned by Eq. (2) means a pair of nodes are very likely to belong to the same cluster. If $\phi(\mathbf{C})_{ij} \doteq \sum_j \mathbf{V}\mathbf{V}_{ij}^T \mathbf{A}_{ij}$, we have:

$$\mathbf{C}_{ij} = \underset{\mathbf{V}\mathbf{V}_{ij}^T}{\arg\min} (\mathbf{A}_{ij} - \sum_j \mathbf{V}\mathbf{V}_{ij}^T \mathbf{A}_{ij})^2. \tag{3}$$

As shown in Eq. (3), $\mathbf{C}_{ij}$ is the coefficient of self-expressiveness [10] (SC) which may describe the global relation between node $i$ and $j$. A higher $\mathbf{C}_{ij}$ inferred by Eq. (3) means the global structure

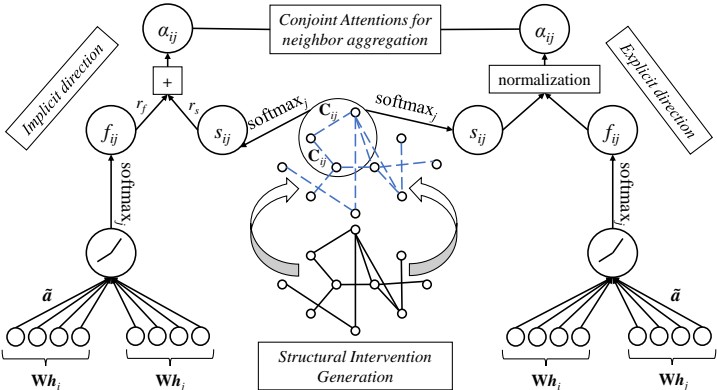

Figure 1: Graphical illustration of the Conjoint Attention layer used in CATs. Left: CA mechanism using *Implicit direction* strategy (CAT-I). Right: CA mechanism using *Explicit direction* strategy (CAT-E). Both two mechanisms consider learnable structural interventions.

of node $i$ can be better represented by that of node $j$, and consequently this pair of nodes are more structurally correlated. It is known that both aforementioned properties have not been considered previously by empirical graph attention mechanisms. We believe considering either of them as structural interventions for the Conjoint Attentions could lead to better attention scores for feature aggregation. Note that other types of $\mathbf{C}$ may also be feasible for the proposed attention mechanisms, as long as they are able to capture meaningful property which is not already possessed within the node embeddings of the GNN.

### 3.3 Conjoint Attention layer

Having obtained a proper $\mathbf{C}$, we present next the Conjoint Attention layer, which is the core module for building CATs and will be used in our experimental study. Different from the attention layers considered in other GNNs, the attention layer proposed here adopts novel attention mechanisms, i.e., Conjoint Attentions. It is known that empirical graph attentions solely concern the correlations pertaining to the internal factors, e.g., node embeddings locating in each layer of the neural net. How those diverse forms of relevance, e.g., the correlation in terms of node cluster embeddings and self-expressiveness may affect the representation learning is therefore yet to be investigated in previous works. Besides utilizing the correlations pertaining to node embeddings, the proposed Conjoint Attentions now take into additional considerations the structural interventions brought by diverse node-node relevance, which is learned outside of the neural network. As a result, each Conjoint Attention layer may pay more attentions to the similar embeddings of neighbors, as well as to the ones that share other forms of relevance with the central node. Node representations possessing heterogeneous forms of relevance can now be learned by the CATs.

Given a set of node features $\{\mathbf{h}_i^l\}_{i=1,\dots N}$, each $\mathbf{h}_i^l \in R^{D^l}$, the Conjoint Attention layer maps them into $D^{l+1}$ dimensional space $\{\mathbf{h}_i^{l+1}\}_{i=1,\dots N}$, according to the correlations of node features and aforementioned structural interventions. The contextual correlation between two connected nodes, say $v_i$ and $v_j$ is firstly obtained. To do so, we directly adopt the feature-based attention mechanism considered by existing graph attention networks (GAT) [33]:

$$f_{ij} = \frac{\exp(\text{LeakyReLU}(\tilde{\mathbf{a}}^T(\mathbf{W}^l\mathbf{h}_i^l \parallel \mathbf{W}^l\mathbf{h}_j^l)))}{\sum_{k\in\mathcal{N}_i}\exp(\text{LeakyReLU}(\tilde{\mathbf{a}}^T(\mathbf{W}^l\mathbf{h}_i^l \parallel \mathbf{W}^l\mathbf{h}_k^l)))}, \qquad (4)$$

where $\tilde{\mathbf{a}} \in \mathbb{R}^{2D^{l+1}}$ is a vector of parameters of the feedforward layer, $\parallel$ stands for the concatenation function, and $\mathbf{W}^l$ is $D^{l+1} \times D^l$ parameter matrix for feature mapping. Given Eq. (4), the proposed CA layer captures the feature correlations between connected nodes (first-order neighbors) by computing the similarities w.r.t. node features mapped to next layer.

As mentioned, determining the attention scores solely based on node features internal to a GNN may result in overlooking other important factors. To overcome this issue, the proposed CA layer attempts to learn the structural interventions as described in Section 3.2, and presents the additional

new information for computing the attention scores. Given any learnable parameter $\mathbf{C}_{ij}$ (structural intervention) between two nodes, CA layer additionally obtains a supplementary correlation as follows:

$$s_{ij} = \frac{\exp\left(\mathbf{C}_{ij}\right)}{\sum_{k \in \mathcal{N}_i} \exp\left(\mathbf{C}_{ik}\right)}. \tag{5}$$

Given $f_{ij}$ and $s_{ij}$, we propose two different strategies to compute the Conjoint Attention scores, aiming at allowing CATs to depend on the structural intervention at different levels. The first mechanism is referred here as *Implicit direction*. It aims at computing the attention scores whose relative significance between structural and feature correlations can be automatically acquired. To do so, each CA layer introduces two learnable parameters, $g_f$ and $g_s$, to determine the relative significance between feature and structural correlations and they can be obtained as follows:

$$r_f = \frac{\exp\left(g_f\right)}{\exp\left(g_s\right) + \exp\left(g_f\right)}, r_s = \frac{\exp\left(g_s\right)}{\exp\left(g_s\right) + \exp\left(g_f\right)}, \tag{6}$$

where $r_s$ or $r_f$ represents the normalized significance related to different types of correlations. Given them, CAT then computes the attention score based on *Implicit direction* strategy:

$$\alpha_{ij} = \frac{r_f \cdot f_{ij} + r_s \cdot s_{ij}}{\sum_{k \in \mathcal{N}_i} [r_f \cdot f_{ik} + r_s \cdot s_{ik}]} = r_f \cdot f_{ij} + r_s \cdot s_{ij}. \tag{7}$$

Given the attention mechanism shown in Eq. (7), $\alpha_{ij}$ attempts to capture the weighted mean attention in terms of the various node-node correlations, which are the ones internal or external to the GNN. Compared with the attention mechanism solely based on features of one-hop neighbors, $\alpha_{ij}$ computed by Eq. (7) may be adapted according to the implicit impact brought by different structural interventions, e.g., correlations pertaining to node cluster embeddings and self-expressiveness coefficients. Moreover, the relative significance $r$ can also be automatically inferred through the back propagation process. More smooth and appropriate attention scores can thereby be computed by the CA layer for learning meaningful representations.

To enhance the impact of structural intervention, the CA layer has another strategy, named here as *Explicit direction*, to compute attention scores between neighbors. Given $f_{ij}$ and $s_{ij}$, the attention scores obtained via the *Explicit direction* strategy is defined as follows:

$$\alpha_{ij} = \frac{f_{ij} \cdot s_{ij}}{\sum_{k \in \mathcal{N}_i} f_{ik} \cdot s_{ik}}. \tag{8}$$

Compared with Eq. (7), $s_{ij}$ explicitly influences the magnitude of $f_{ij}$, so that those node pairs which are irrelevant in terms of $\mathbf{C}_{ij}$ will never be assigned with high attention weights. Based on *Explicit direction* strategy, the CA layer becomes more structurally dependent when performing message passing to the higher layers in the neural architecture.

Having obtained the Conjoint Attention scores, the CA layer is now able to compute a linear combination of features corresponding to each node and its neighbors as output, which will be either propagated to the higher layer, or be used as the final representations for subsequent learning tasks. The described output features can be computed as follows:

$$\mathbf{h}_i^{l+1} = (\alpha_{ii} + \epsilon \cdot \frac{1}{|\mathcal{N}_i|})\mathbf{W}^l\mathbf{h}_i^l + \sum_{j \in \mathcal{N}_i, j \neq i} \alpha_{ij}\mathbf{W}^l\mathbf{h}_j^l, \tag{9}$$

where $\epsilon \in (0, 1)$ is a learnable parameter that improves the expressive capability of the proposed CA layer.

### 3.4 Construction of Graph conjoint attention networks (CATs)

In Fig. 1, the Conjoint Attention layers that use the attention mechanisms proposed in this paper are graphically illustrated. We are now able to construct Graph conjoint attention networks (CATs) using a particular number of CA layers proposed. In practice, we also adopt the multi-head attention strategy [32] to stabilize the learning process. CATs may either concatenate the node features from multiple hidden layers as the input for next layer, or compute the average of node features obtained by multiple units of output layers as the final node representations. For the details on implementing multi-head attention in graph neural networks, the reader is referred to [33].

# 4 Theoretical analysis

Study on the expressive power of various GNNs has drawn much attention in the recent. It concerns whether a given GNN can distinguish different structures where vertices possessing various vectorized features. It has been found that what the neighborhood aggregation functions of all message-passing GNNs aim at are analogous to the 1-dimensional Weisfeiler-Lehman test (1-WL test), which is injective and iteratively operated in the Weisfeiler-Lehman algorithm [36, 41, 45], does. As a result, all message-passing GNNs are as most powerful as the 1-WL test [41]. The theoretical validation of the expressive power of a given GNN thereby lies in whether those adopted aggregation/readout functions are homogeneous to the 1-WL test.

One may naturally be interested in whether the expressive power of the proposed CAT layers is as powerful as the 1-WL test, which can distinguish all different graph structures. To do so, we firstly show that neighborhood aggregation function (Eq. (9)) without the term for improving expressive capability (i.e., $\epsilon \cdot \frac{1}{|\mathcal{N}_i|} \mathbf{W} \mathbf{h}_i^l$ in Eq. (9)) still fails to discriminate some graph structures possessing certain topological properties. Then, by integrating the term of improving expressive capability, all the proposed CA layers are able to distinguish all those graph structures that cannot be discriminated previously.

For the function of neighborhood aggregation solely utilizing the strategy shown in Eq. (7), we have the following theorem pointing out the conditions under which the aggregation function fails to distinguish different structures.

**Theorem 1** *Assume the feature space $\mathcal{X}$ is countable and the aggregation function using the weights computed by Eq. (7) is represented as $h(c, X) = \sum_{x \in X} \alpha_{cx} g(x)$, where $c$ is the feature of center node, $X \in \mathcal{X}$ is a multiset containing the feature vectors from nodes in $\mathcal{N}_i$, $g(\cdot)$ is a function for mapping input feature $X$, and $\alpha_{cx}$ is the weight between $g(c)$ and $g(x)$. For all $g$ and the strategy in Eq. (7), $h(c_1, X_1) = h(c_2, X_2)$ if and only if $c_1 = c_2$, $X_1 = \{S, \mu_1\}$, $X_2 = \{S, \mu_2\}$, and $\sum_{y=x, y \in X_1} f_{c_1 y} - \sum_{y=x, y \in X_2} f_{c_2 y} = q[\sum_{y=x, y \in X_2} s_{c_2 y} - \sum_{y=x, y \in X_1} s_{c_1 y}]$, for $q = \frac{r_s}{r_f}$ and $x \in S$. In other words, $h$ will map different multisets into the same embedding iff the multisets have same central node feature, same underlying set, and the difference in feature-based scores is proportional ($\frac{r_s}{r_f}$) to the opposite of that in the weights corresponding to the structural interventions.*

We leave the proof of all the theorems and corollaries in the appendix. For the aggregation function utilizing the strategy shown in Eq. (8), we have the following theorem indicating the structures which cannot be correctly distinguished.

**Theorem 2** *Under the same assumptions shown in Theorem 1, for all $g$ and the strategy in Eq. (8), $h(c_1, X_1) = h(c_2, X_2)$ if and only if $c_1 = c_2$, $X_1 = \{S, \mu_1\}$, $X_2 = \{S, \mu_2\}$, and $q \cdot \sum_{y=x, y \in X_1} \phi(\mathbf{C}_{c_1 x}) = \sum_{y=x, y \in X_2} \phi(\mathbf{C}_{c_2 y})$, for $q > 0$ and $x \in S$, where $\phi(\cdot)$ is an function for mapping values to $\mathbb{R}^+$. In other words, $h$ will map different multisets into the same embedding iff the multisets have same central node feature, same node features whose corresponding mapped structural interventions are proportional.*

Theorems 1 and 2 indicate that the CA layers may still fail to distinguish some structures, if they exclude the improving term shown in Eq. (9). However, GNNs utilizing Eqs. (7) or (8) can still be more expressively powerful than classical GATs. As node features and structural interventions are heterogeneous, intuitively, structures satisfying the stated conditions should be infrequent. This may well explain why those GNNs concerning including external factors, e.g., some structural properties into the computation of attention coefficients may experimentally perform better than GATs. However, when distinct multisets with corresponding properties meet the conditions mentioned in Theorems 1 and 2, the attention mechanisms solely based on Eqs. (7) or (8) cannot correctly distinguish such multisets. Thus, GNNs only utilizing Eqs. (7) or (8) as the feature aggregation function fail to reach the upper bound of expressive power of all message-passing GNNs, i.e., the 1-WL test.

However, we are able to readily improve the expressive power of CATs to meet the condition of the 1-WL test by slightly modifying the aggregation function as Eq. (9) shows. Then, the newly obtained Conjoint Attention scores can be used to aggregate the node features passed to the higher layers. Next, we prove that the proposed Conjoint Attention mechanisms (Eqs. (7)-(9)) reach the upper bound of message-passing GNNs via showing they can distinguish those structures possessing the properties mentioned in Theorems 1 and 2.

**Corollary 1** *Let $\mathcal{T}$ be the attention-based aggregator shown in Eq. (9) that considers one of the strategies in Eq. (7) or (8) and operates on a multiset $H \in \mathcal{H}$, where $\mathcal{H}$ is a node feature space mapped from the countable input feature space $\mathcal{X}$. A $\mathcal{H}$ exists so that utilizing attention-based aggregator shown in Eq. (9), $\mathcal{T}$ can distinguish all different multisets in aggregation that it previously cannot discriminate.*

Based on the performed analysis, the expressive power of CATs is theoretically stronger than state-of-the-art attention-based GNNs, e.g., GATs [33].

## 5 Experiments and analysis

In this section, we evaluate the proposed Graph conjoint attention networks against a variety of state-of-the-art and popular baselines, on widely used network datasets.

### 5.1 Experimental set-up

**Baselines for comparison**-To validate the effectiveness of the proposed CATs, we compare them with a number of state-of-the-art baselines, including Arma filter GNN (ARMA) [2], Simplified graph convolutional Networks (SGC) [37], Personalized Pagerank GNN (APPNP) [18], Graph attention networks (GAT) [33], Jumping knowledge networks (JKNet) [42], Graph convolutional networks (GCN) [17], GraphSAGE [13], Mixture model CNN (MoNet) [23], and Graph isomorphism network (GIN) [41]. As GAT can alternatively consider graph structure by augmenting original node features (i.e., $\mathbf{X}$) with structural properties, we use prevalent methods for network embedding, including $k$-eigenvectors of graph Laplacian ($k$-Lap) [26], Deepwalk [25], and Matrix factorization-based network embedding (NetMF) [26] to learn structural node representations and concatenate them with $\mathbf{X}$ as the input feature of GAT. Thus, three variants of GAT, i.e., GAT-$k$-Lap, GAT-Deep, and GAT-NetMF are additionally constructed as compared baselines. Based on the experimental results previously reported, these baselines may represent the most advanced techniques for learning in graph structured data.

**Testing datasets**-Five widely-used network datasets, which are Cora, Cite, Pubmed [21, 28], CoauthorCS [29], and OGB-Arxiv [15], are used in our experiments. Cora, Cite, and Pubmed are three classical network datasets for validating the effectiveness of GNNs. However, it is recently found that these three datasets sometimes may not effectively validate the predictive power of different graph learning approaches, due to the relatively small data size and data leakage [15, 29]. Thus, more massive datasets having better data quality have been proposed to evaluate the performance of different approaches [9, 15]. In our experiment, we additionally use CoauthorCS and OGB-Arxiv as testing datasets. The details of all benchmarking sets can be checked in the appendix.

**Evaluation and experimental settings**-Two learning tasks, semi-supervised node classification and semi-supervised node clustering are considered in our experiments. For the training paradigms of both two learning tasks, we closely follow the experimental scenarios established in the related works [15, 17, 33, 43]. For the testing phase of different approaches, we use the test splits that are publicly available for classification tasks, and all nodes for clustering tasks. The effectiveness of all methods is validated through evaluating the classified nodes using $Accuracy$. In the training stage, we construct the two-layer network structure (i.e., one hidden layer possessed) for all the baselines and different versions of CATs. In each set of testing data, all approaches are run ten times to obtain the statistically steady performance. As for other details related to experimental settings, we leave them in the appendix.

### 5.2 Results on node classification

The results on semi-supervised node classification are summarized in Table 1. As the table shows, CATs utilizing different attention strategies generally perform better than any other baseline in all the testing datasets. Specifically, CAT utilizing *Implicit direction* (CAT-I-MF and CAT-I-SC) performs better than all the compared baselines in all the five datasets. CAT utilizing *Explicit direction* (CAT-E-MF and CAT-E-SC) is better than other compared baselines in four datasets out of five, except the case of CoauthorCS. In that dataset, CAT-E ranks the second-best when compared with other baselines.

Table 1: Average *Accuracy* on semi-supervised node classification. Bold fonts mean CAT obtains a better performance than any other baseline.

| | Cora | Cite | Pubmed | CoauthorCS | OGB-Arxiv |
|---|---|---|---|---|---|
| MoNet | $81.96 \pm 0.50$ | $64.22 \pm 0.16$ | $79.78 \pm 0.33$ | $91.96 \pm 0.75$ | $47.71 \pm 0.27$ |
| GCN | $81.42 \pm 0.19$ | $71.60 \pm 0.73$ | $79.66 \pm 0.39$ | $91.54 \pm 0.43$ | $71.78 \pm 0.16$ |
| GraphSAGE | $81.12 \pm 0.41$ | $71.06 \pm 0.64$ | $79.04 \pm 0.62$ | $93.06 \pm 0.80$ | $69.07 \pm 0.27$ |
| JKNet | $78.34 \pm 0.02$ | $65.88 \pm 0.01$ | $79.88 \pm 0.01$ | $89.62 \pm 0.01$ | $64.91 \pm 0.01$ |
| APPNP | $82.80 \pm 0.32$ | $72.38 \pm 0.50$ | $82.62 \pm 0.37$ | $89.16 \pm 0.65$ | $63.16 \pm 0.54$ |
| SGC | $81.90 \pm 0.01$ | $71.40 \pm 0.01$ | $82.42 \pm 0.04$ | $93.60 \pm 0.01$ | $61.06 \pm 0.09$ |
| ARMA | $80.06 \pm 0.57$ | $70.00 \pm 0.66$ | $76.46 \pm 0.58$ | $86.28 \pm 0.75$ | $68.77 \pm 0.17$ |
| GIN | $81.58 \pm 0.62$ | $66.90 \pm 0.16$ | $80.76 \pm 0.33$ | $93.03 \pm 0.74$ | $64.02 \pm 0.18$ |
| GAT | $83.84 \pm 0.61$ | $70.36 \pm 0.42$ | $81.50 \pm 0.47$ | $92.80 \pm 0.41$ | $72.39 \pm 0.07$ |
| GAT-$k$-Lap | $84.10 \pm 0.24$ | $71.18 \pm 0.52$ | $82.56 \pm 0.30$ | $92.70 \pm 0.31$ | $72.47 \pm 0.06$ |
| GAT-NetMF | $84.44 \pm 0.19$ | $70.94 \pm 0.16$ | $81.90 \pm 0.33$ | $93.16 \pm 0.27$ | $72.42 \pm 0.08$ |
| GAT-Deep | $83.68 \pm 0.67$ | $69.70 \pm 0.57$ | $80.13 \pm 0.26$ | $92.93 \pm 0.17$ | $72.79 \pm 0.09$ |
| CAT-I-MF | $\mathbf{85.38 \pm 0.16}$ | $\mathbf{73.22 \pm 0.19}$ | $\mathbf{83.90 \pm 0.24}$ | $\mathbf{93.74 \pm 0.14}$ | $\mathbf{72.89 \pm 0.06}$ |
| CAT-I-SC | $85.50 \pm 0.22$ | $73.18 \pm 0.22$ | $84.28 \pm 0.20$ | $93.70 \pm 0.11$ | $72.85 \pm 0.04$ |
| CAT-E-MF | $85.56 \pm 0.19$ | $73.24 \pm 0.21$ | $83.60 \pm 0.17$ | $93.40 \pm 0.12$ | $72.81 \pm 0.09$ |
| CAT-E-SC | $\mathbf{85.40 \pm 0.36}$ | $\mathbf{73.02 \pm 0.24}$ | $\mathbf{84.02 \pm 0.24}$ | $93.30 \pm 0.11$ | $\mathbf{72.83 \pm 0.11}$ |

Table 2: Average *Accuracy* on semi-supervised node clustering. Bold fonts mean CAT obtains a better performance than any other baseline.

| | Cora | Cite | Pubmed | CoauthorCS | OGB-Arxiv |
|---|---|---|---|---|---|
| MoNet | $79.42 \pm 0.86$ | $63.07 \pm 0.11$ | $79.39 \pm 0.61$ | $88.75 \pm 0.54$ | $53.08 \pm 0.15$ |
| GCN | $74.25 \pm 0.13$ | $63.36 \pm 0.87$ | $77.83 \pm 0.75$ | $89.74 \pm 0.53$ | $75.02 \pm 0.07$ |
| GraphSAGE | $78.46 \pm 0.56$ | $69.00 \pm 0.17$ | $79.52 \pm 1.13$ | $90.16 \pm 0.53$ | $73.50 \pm 0.13$ |
| JKNet | $75.95 \pm 0.01$ | $65.12 \pm 0.03$ | $79.52 \pm 0.01$ | $86.66 \pm 0.01$ | $71.28 \pm 0.01$ |
| APPNP | $79.93 \pm 0.82$ | $70.55 \pm 0.85$ | $82.81 \pm 0.32$ | $85.93 \pm 0.39$ | $69.73 \pm 0.67$ |
| SGC | $79.38 \pm 0.02$ | $69.71 \pm 0.02$ | $81.64 \pm 0.01$ | $90.13 \pm 0.01$ | $71.09 \pm 0.37$ |
| ARMA | $77.70 \pm 0.99$ | $68.38 \pm 0.87$ | $77.29 \pm 1.11$ | $84.72 \pm 0.29$ | $69.24 \pm 0.12$ |
| GIN | $78.25 \pm 0.46$ | $67.83 \pm 0.15$ | $79.31 \pm 0.35$ | $89.97 \pm 0.26$ | $63.85 \pm 0.18$ |
| GAT | $81.39 \pm 0.18$ | $69.20 \pm 0.28$ | $80.88 \pm 0.33$ | $90.09 \pm 0.15$ | $76.04 \pm 0.38$ |
| GAT-$k$-Lap | $80.66 \pm 0.31$ | $69.56 \pm 0.34$ | $81.59 \pm 0.09$ | $89.83 \pm 0.18$ | $76.21 \pm 0.06$ |
| GAT-NetMF | $81.75 \pm 0.26$ | $68.96 \pm 0.21$ | $81.74 \pm 0.18$ | $89.85 \pm 0.21$ | $76.06 \pm 0.07$ |
| GAT-Deep | $81.08 \pm 0.41$ | $68.27 \pm 0.06$ | $80.55 \pm 0.11$ | $89.70 \pm 0.27$ | $76.91 \pm 0.15$ |
| CAT-I-MF | $\mathbf{82.17 \pm 0.11}$ | $\mathbf{71.15 \pm 0.12}$ | $82.77 \pm 0.07$ | $\mathbf{90.26 \pm 0.22}$ | $\mathbf{77.72 \pm 0.07}$ |
| CAT-I-SC | $82.26 \pm 0.13$ | $71.17 \pm 0.15$ | $82.86 \pm 0.07$ | $90.29 \pm 0.21$ | $77.01 \pm 0.16$ |
| CAT-E-MF | $\mathbf{81.98 \pm 0.19}$ | $\mathbf{71.21 \pm 0.12}$ | $82.40 \pm 0.08$ | $89.66 \pm 0.22$ | $\mathbf{76.93 \pm 0.16}$ |
| CAT-E-SC | $\mathbf{82.01 \pm 0.24}$ | $\mathbf{71.11 \pm 0.24}$ | $82.61 \pm 0.14$ | $89.72 \pm 0.15$ | $\mathbf{76.98 \pm 0.08}$ |

## 5.3 Results on node clustering

Node clustering can be more challenging as all the nodes containing various potential structures in the graph are used in the testing phase. The results obtained show that CATs still performs robustly when compared with other baselines on this challenging task. As Table 2 shows, the *Implicit direction* strategy utilized by CAT can still ensure the proposed neural architecture to outperform other compared baselines in all the datasets. As for CAT utilizing *Explicit direction*, it ranks best on three datasets out of five. While on the remaining two datasets, Pubmed and CoauthorCS, the performance of CAT-E is competitive to the best. Based on the robust performance shown in Tables 1 and 2, CAT is observed to be one of the most effective GNNs for various graph learning tasks.

## 5.4 Ablation study

To further investigate whether the proposed CA mechanisms are effective in improving the predictive power of CATs, we compared the performance of CAs with that obtained by attention mechanisms considering various factors. Specifically, we let GAT computes attention coefficients using either

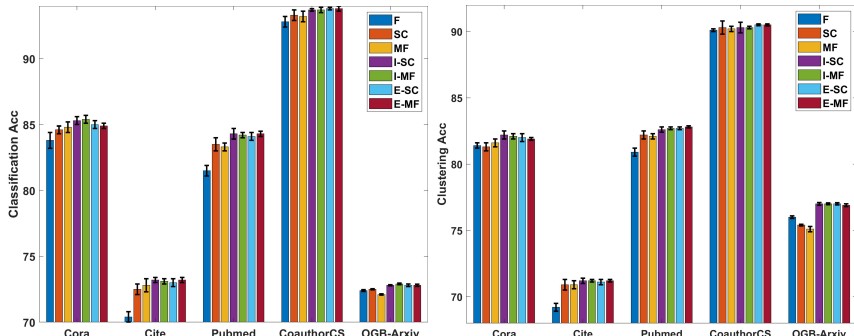

Figure 2: Performance comparison on computing attention scores using different factors

structural interventions, i.e., node-node correlations pertaining to node cluster embeddings (MF in Eq. (2)) and self-representation coefficients (SC in Eq. (3)), or node features (Eq.(4), F). Then GAT utilizing different attention strategies is used to perform node classification and clustering tasks on all the testing datasets. The performance comparisons between CATs and GAT utilizing the aforementioned attentions have been summarized in Fig. 2. As the figure shows, on both classification and clustering tasks, the proposed CA mechanisms perform statistically better than other attention strategies utilized by GAT. It is also observed that the consideration of structural attentions (SC and MF) may also improve the performance of attention-based GNNs. Such results thus experimentally agree with and validates our theoretical analysis in Section 4.

## 6  Discussions

In this section, some further discussions that may provide better understandings to the proposed attention mechanisms are presented.

### 6.1  Comparisons between CATs and GATs with augmented node features

Besides the proposed Conjoint Attention mechanisms, directly concatenating input node features and structural embeddings [11, 25–27, 31] is another effective way to make current GNNs more structural bias. In our experiments, it is shown that the learning performance have improved in most datasets when GAT uses the concatenation of original node features and various structural embeddings. However, in most datasets, such performance improvement is not as significant as that obtained by CATs. Different from directly concatenating the original node input features and structural embeddings for learning node representations, CATs provide a smooth means to compute attention coefficients that jointly consider the diverse relevance regarding to both layer-wise node embeddings and external factors, like the structural interventions pertaining to the correlations generated by the node cluster embeddings considered in this paper. As a result, CATs can learn node representations from those nodes that are heterogeneously relevant and attain notable learning performance on all the testing datasets.

### 6.2  Potential limitations of the proposed approach

Although the proposed Conjoint Attentions are very effective in improving the learning performance of attention-based GNNs, they may also have possible shortcomings. First, the predictive power of the proposed CATs may be determined by the quality of the structural interventions. As the proposed Conjoint Attentions attempt to compute attention scores considering heterogeneous factors, their performance might be negatively affected by those contaminated ones. However, some potential methods may mitigate the side-effect brought by possible false/noisy external factors. One is to utilize adversarial learning modules [20] to alleviate the model vulnerability to external contamination. The other method is to consider learning effective factors from multiple data sources (i.e., multi-view). In previous works [39], multi-view learning has shown to be effective even when some view of data is contaminated.

Second, both space and time complexity of the proposed Conjoint Attentions can be higher than the empirical attention-based GNNs, e.g., GAT. Thus, how to select a simple but effective strategy for capturing the interventions is crucial for the proposed Conjoint Attentions. In our experiments, we recorded the memory consumption of CATs when performing different learning tasks in all the testing datasets and the corresponding results are provided in the appendix. We find that some simple learning paradigms, e.g., matrix factorization (Eq. (2) in the manuscript) can ensure CATs to outperform other baselines, while the space and time complexity does not increase much when compared with GATs. As for the SC strategy, its space complexity is relatively high, if it uses a single-batch optimization method. Therefore, more efficient optimization methods should be considered when CATs use SC to learn $\mathbf{C}_{ij}$.

Third, the expressive power of the proposed Conjoint Attention-based GNNs reaches the upper bound of the 1-WL test in the countable feature space, but such discriminative capacity may not be always held by CATs in the uncountable feature space. Previous works have proved that the expressive power of a simplex function for feature aggregation in a GNN can surely reach the upper bound of the 1-WL test only in countable feature space, multiple categories of functions for feature aggregations are required to maintain the expressive power of a GNN when the feature space is uncountable [6]. As all Conjoint Attentions belong to one category of function, i.e., mean aggregator, in this paper, we perform the theoretical analysis on the expressive power of CATs assuming the feature space is countable. Ideally, the expressive power of CATs can be further improved in uncountable space, if the proposed Conjoint Attentions are appropriately combined with other types of feature aggregators.

## 7   Conclusion

In this paper, we have proposed a class of novel attention strategies, known as Conjoint Attentions (CAs) to construct Graph conjoint attention networks (CATs). Different from empirical graph attentions, CAs offer flexible incorporation of both layer-wise node features and structural interventions that can be learned outside of the GNN to compute appropriate weights for feature aggregation. Besides, the expressive power of CATs is theoretically validated to reach the upper bound of all message-passing GNNs. The proposed CATs have been compared with a number of prevalent approaches in different learning tasks. The obtained notable results verify the CATs' model effectiveness. In future, we will further improve the effectiveness of CATs in the following ways. First, besides node cluster embeddings and self-expressiveness, more structural interventions will be explored to compute more compelling attention coefficients for node representation learning. Second, appropriate adversarial strategies to reduce model vulnerability to contaminated factors that are used for computing attention scores will be considered. Last but not the least, the proposed Conjoint Attentions will be extended to learn node representations in multi-view context and heterogeneous graph data.

## Acknowledgments and Disclosure of Funding

The authors would like to thank the anonymous reviewers for their constructive comments and suggestions. This work is supported in part by the Data Science & Artificial Intelligence Research Center (DSAIR), Nanyang Technological University, and in part by Agency for Science, Technology and Research (A*STAR).

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
