# Learning Conjoint Attentions for Graph Neural Nets
## Supplementary Materials

**Tiantian He[1,2]  Yew-Soon Ong[1,2]  Lu Bai[1,2]**
[1]Agency for Science, Technology and Research (A*STAR)
[2]DSAIR, Nanyang Technological University
{He_Tiantian,Bai_Lu}@ihpc.a-star.edu.sg, Ong_Yew_Soon@hq.a-star.edu.sg
{tiantian.he,bailu,asysong}@ntu.edu.sg

## A  Proof of Theorem 1

To prove Theorem 1, we need to consider the two directions of the iff conditions. If given $c_1 = c_2$, $S_1 = S_2$, and $\sum_{y=x,y\in X_1} f_{c_1 y} - \sum_{y=x,y\in X_2} f_{c_2 y} = q[\sum_{y=x,y\in X_2} s_{c_2 y} - \sum_{y=x,y\in X_1} s_{c_1 y}]$, for $q = \frac{r_s}{r_f}$, for the aggregation function utilizing the weights computed by Eq. (7) in the manuscript, we have:

$$h(c_i, X_i) = \sum_{x \in X_i} \alpha_{c_i x} g(x),$$
$$\alpha_{c_i x} = r_f \cdot f_{c_i x} + r_s \cdot s_{c_i x}, \tag{1}$$
$$f_{c_i x} = \frac{\exp(m_{c_i x})}{\sum_{x \in X_i} \exp(m_{c_i x})}, s_{c_i x} = \frac{\exp(\mathbf{C}_{c_i x})}{\sum_{x \in X_i} \exp(\mathbf{C}_{c_i x})},$$

where $m_{c_i x}$ represents the feature similarity between $c_i$ and $x$. Given Eq. (1), we may directly derive $h(c_1, X_1)$ and $h(c_2, X_2)$:

$$h(c_1, X_1) = \sum_{x \in X_1} \alpha_{c_1 x} g(x) = \sum_{x \in X_1} [r_f \cdot f_{c_1 x} + r_s \cdot s_{c_1 x}] \cdot g(x)$$
$$h(c_2, X_2) = \sum_{x \in X_2} \alpha_{c_2 x} g(x) = \sum_{x \in X_2} [r_f \cdot f_{c_2 x} + r_s \cdot s_{c_2 x}] \cdot g(x) \tag{2}$$

Considering $c_1 = c_2$, $S_1 = S_2$, and $\sum_{y=x,y\in X_1} f_{c_1 y} - \sum_{y=x,y\in X_2} f_{c_2 y} = q[\sum_{y=x,y\in X_2} s_{c_2 y} - \sum_{y=x,y\in X_1} s_{c_1 y}]$, for $q = \frac{r_s}{r_f}$, we directly derive $h(c_1, X_1) = h(c_2, X_2)$.

If we are given $h(c_1, X_1) = h(c_2, X_2)$, we are able to prove that the conditions mentioned in the theorem are necessary by showing contradictions occur when they are not satisfied. If $h(c_1, X_1) = h(c_2, X_2)$, we have:

$$h(c_1, X_1) - h(c_2, X_2) =$$
$$\sum_{x \in X_1} [r_f \cdot f_{c_1 x} + r_s \cdot s_{c_1 x}] \cdot g(x) - \sum_{x \in X_2} [r_f \cdot f_{c_2 x} + r_s \cdot s_{c_2 x}] \cdot g(x) = 0 \tag{3}$$

35th Conference on Neural Information Processing Systems (NeurIPS 2021).

Firstly, assuming $S_1 \neq S_2$, for any $g(\cdot)$, we thereby have:

$$
\begin{aligned}
h(c_1, X_1) - h(c_2, X_2) = & \sum_{x \in S_1 \cap S_2} [ \sum_{y=x, y \in X_1} [r_f \cdot f_{c_1 y} + r_s \cdot s_{c_1 y}] \\
& - \sum_{y=x, y \in X_2} [r_f \cdot f_{c_2 y} + r_s \cdot s_{c_2 y}]] \cdot g(x) \\
& + \sum_{x \in S_1 \setminus S_2} \sum_{y=x, y \in X_1} [r_f \cdot f_{c_1 y} + r_s \cdot s_{c_1 y}] \cdot g(x) \\
& - \sum_{x \in S_2 \setminus S_1} \sum_{y=x, y \in X_2} [r_f \cdot f_{c_2 y} + r_s \cdot s_{c_2 y}] \cdot g(x) = 0
\end{aligned}
\tag{4}
$$

As Eq. (4) holds for any $g(\cdot)$, we may define another function $g'(\cdot)$ as follows:

$$
\begin{aligned}
g(x) &= g'(x), \text{ for } x \in S_1 \cap S_2 \\
g(x) &= g'(x) - 1, \text{ for } x \in S_1 \setminus S_2 \\
g(x) &= g'(x) + 1, \text{ for } x \in S_2 \setminus S_1
\end{aligned}
\tag{5}
$$

If Eq. (4) holds, we also have:

$$
\begin{aligned}
h(c_1, X_1) - h(c_2, X_2) = & \sum_{x \in S_1 \cap S_2} [ \sum_{y=x, y \in X_1} [r_f \cdot f_{c_1 y} + r_s \cdot s_{c_1 y}] \\
& - \sum_{y=x, y \in X_2} [r_f \cdot f_{c_2 y} + r_s \cdot s_{c_2 y}]] \cdot g'(x) \\
& + \sum_{x \in S_1 \setminus S_2} \sum_{y=x, y \in X_1} [r_f \cdot f_{c_1 y} + r_s \cdot s_{c_1 y}] \cdot g'(x) \\
& - \sum_{x \in S_2 \setminus S_1} \sum_{y=x, y \in X_2} [r_f \cdot f_{c_2 y} + r_s \cdot s_{c_2 y}] \cdot g'(x) \\
= & \sum_{x \in S_1 \cap S_2} [ \sum_{y=x, y \in X_1} [r_f \cdot f_{c_1 y} + r_s \cdot s_{c_1 y}] \\
& - \sum_{y=x, y \in X_2} [r_f \cdot f_{c_2 y} + r_s \cdot s_{c_2 y}]] \cdot g(x) \\
& + \sum_{x \in S_1 \setminus S_2} \sum_{y=x, y \in X_1} [r_f \cdot f_{c_1 y} + r_s \cdot s_{c_1 y}] \cdot [g(x) + 1] \\
& - \sum_{x \in S_2 \setminus S_1} \sum_{y=x, y \in X_2} [r_f \cdot f_{c_2 y} + r_s \cdot s_{c_2 y}] \cdot [g(x) - 1] = 0
\end{aligned}
\tag{6}
$$

As Eq. (4) equals Eq. (6), we have:

$$
\sum_{x \in S_1 \setminus S_2} \sum_{y=x, y \in X_1} [r_f \cdot f_{c_1 y} + r_s \cdot s_{c_1 y}] + \sum_{x \in S_2 \setminus S_1} \sum_{y=x, y \in X_2} [r_f \cdot f_{c_2 y} + r_s \cdot s_{c_2 y}] = 0
\tag{7}
$$

Obviously, the above equation does not hold as the terms in the summation operator are positive. Thus, $S_1 \neq S_2$ is not true. We may now assume $S_1 = S_2 = S$. Eliminating the irrational terms in Eq. (4), we have:

$$
\sum_{x \in S_1 \cap S_2} [ \sum_{y=x, y \in X_1} [r_f \cdot f_{c_1 y} + r_s \cdot s_{c_1 y}] - \sum_{y=x, y \in X_2} [r_f \cdot f_{c_2 y} + r_s \cdot s_{c_2 y}]] \cdot g(x) = 0
\tag{8}
$$

Thus we know each term in the summation equals zero:

$$
\sum_{y=x, y \in X_1} [r_f \cdot f_{c_1 y} + r_s \cdot s_{c_1 y}] - \sum_{y=x, y \in X_2} [r_f \cdot f_{c_2 y} + r_s \cdot s_{c_2 y}] = 0
\tag{9}
$$

As external factors and node feature similarity are heterogeneous, we may assume $\sum_{y=x, y \in X_1} r_s \cdot s_{c_1 y} = \sum_{y=x, y \in X_2} r_s \cdot s_{c_2 y}$. Eq. (9) can be simplified and rewritten as:

$$
\frac{\mu_1(x)}{\mu_2(x)} = \frac{\exp(m_{c_2 x}) \sum_{x \in S_1} \sum_{y=x, y \in X_1} \exp(m_{c_1 x})}{\exp(m_{c_1 x}) \sum_{x \in S_2} \sum_{y=x, y \in X_2} \exp(m_{c_2 x})}
\tag{10}
$$

It is obvious that LHS of Eq. (10) is a rational number. However, the RHS of Eq. (10) can be an irrational number. We may consider $S = \{s, s_0\}$ and assume $c_1 = s_0, c_2 = s$. We may also assume the feature similarity between central node and others as follows:

$$m_{c_1 x} = 1, \text{for } x \in S$$
$$m_{c_2 s} = 1, m_{c_2 s_0} = 2$$
(11)

Consider $x = s$, we have:

$$\frac{\mu_1(s)}{\mu_2(s)} = \frac{|X_1|}{|X_2| - n + ne},$$
(12)

where $n$ stands for the number of $s_0$ in $X_2$. It is obvious that the above equality does not hold as the RHS is an irrational number, while LHS is a rational number. Thus $c_1 \neq c_2$ is false. Given $c_1 = c_2$, Eq. (9) can be rewritten as:

$$\sum_{y=x,y\in X_1} r_f \cdot f_{c_1 y} - \sum_{y=x,y\in X_2} r_f \cdot f_{c_2 y} + \sum_{y=x,y\in X_1} r_s \cdot s_{c_1 y} - \sum_{y=x,y\in X_2} r_s \cdot s_{c_2 y} = 0$$
(13)

To ensure above equation holds, we have:

$$\sum_{y=x,y\in X_1} f_{c_1 y} - \sum_{y=x,y\in X_2} f_{c_2 y} = \frac{r_s}{r_f} [\sum_{y=x,y\in X_2} s_{c_2 y} - \sum_{y=x,y\in X_1} s_{c_1 y}]$$
(14)

Further denoting $\frac{r_s}{r_f} = q$, we have $\sum_{y=x,y\in X_1} f_{c_1 y} - \sum_{y=x,y\in X_2} f_{c_2 y} = q[\sum_{y=x,y\in X_2} s_{c_2 y} - \sum_{y=x,y\in X_1} s_{c_1 y}]$. □

## B   Proof of Theorem 2

To prove Theorem 2, we can follow the procedure which is used to prove Theorem 1. If given $c_1 = c_2$, $S_1 = S_2$, and $q \cdot \sum_{y=x,y\in X_1} \phi(\mathbf{C}_{c_1 y}) = \sum_{y=x,y\in X_1} \phi(\mathbf{C}_{c_2 y})$, for $q > 0$, we may directly replace $\phi(\cdot)$ using $\exp\{\cdot\}$. For the aggregation function solely using the weights obtained by Eq. (8) in the manuscript, we have:

$$h(c_i, X_i) = \sum_{x\in X_i} \alpha_{c_i x} g(x),$$
$$\alpha_{c_i x} = \frac{f_{c_i x} \cdot s_{c_i x}}{\sum_{x\in X_i} f_{c_i x} \cdot s_{c_i x}},$$
$$f_{c_i x} = \frac{\exp(m_{c_i x})}{\sum_{x\in X_i} \exp(m_{c_i x})}, s_{c_i x} = \frac{\exp(\mathbf{C}_{c_i x})}{\sum_{x\in X_i} \exp(\mathbf{C}_{c_i x})},$$
(15)

where $m_{c_i x}$ represents the feature similarity between $c_i$ and $x$. Given Eq. (15), we may directly derive $h(c_1, X_1)$ and $h(c_2, X_2)$:

$$h(c_1, X_1) = \sum_{x\in X_1} \alpha_{c_1 x} g(x) = \sum_{x\in X_1} [\frac{f_{c_1 x} \cdot s_{c_1 x}}{\sum_{x\in X_1} f_{c_1 x} \cdot s_{c_1 x}}] \cdot g(x)$$
$$h(c_2, X_2) = \sum_{x\in X_2} \alpha_{c_2 x} g(x) = \sum_{x\in X_2} [\frac{f_{c_2 x} \cdot s_{c_2 x}}{\sum_{x\in X_2} f_{c_2 x} \cdot s_{c_2 x}}] \cdot g(x)$$
(16)

Given $S_1 = S_2$ and $c_1 = c_2$, $h(c_2, X_2)$ can be rewritten as:

$$h(c_2, X_2) = \sum_{x\in S_2} [\frac{f_{c_2 x} \cdot \sum_{y=x,y\in X_2} s_{c_2 y}}{\sum_{x\in S_2} f_{c_2 x} \cdot \sum_{y=x,y\in X_2} s_{c_2 y}}] \cdot g(x)$$
$$= \sum_{x\in S_2} [\frac{\frac{\exp(m_{c_2 x})}{\sum_{y\in X_2} \exp(m_{c_2 y})} \cdot \sum_{y=x,y\in X_2} \frac{\exp(\mathbf{C}_{c_2 y})}{\sum_{y\in X_2} \exp(\mathbf{C}_{c_2 y})}}{\sum_{x\in S_2} \frac{\exp(m_{c_2 x})}{\sum_{y\in X_2} \exp(m_{c_2 y})} \sum_{y=x,y\in X_2} \frac{\exp(\mathbf{C}_{c_2 y})}{\sum_{y\in X_2} \exp(\mathbf{C}_{c_2 y})}}] \cdot g(x)$$
$$= \sum_{x\in S_2} \frac{\exp(m_{c_2 x}) \sum_{y=x,y\in X_2} \exp(\mathbf{C}_{c_2 y})}{\sum_{x\in S_2} \exp(m_{c_2 x}) \sum_{y=x,y\in X_2} \exp(\mathbf{C}_{c_2 y})} \cdot g(x)$$
(17)

Given $q \cdot \sum_{x \in X_1} \exp\{\mathbf{C}_{c_1 x}\} = \sum_{y=x} \exp\{\mathbf{C}_{c_2 y}\}$, Eq. (17) is equivalent to:

$$h(c_2, X_2) = \sum_{x \in S_1} \frac{q \cdot \exp\left(m_{c_2 x}\right) \sum_{y=x, y \in X_1} \exp\left(\mathbf{C}_{c_1 y}\right)}{\sum_{x \in S_1} q \cdot \exp\left(m_{c_2 x}\right) \sum_{y=x, y \in X_1} \exp\left(\mathbf{C}_{c_1 y}\right)} \cdot g(x) \tag{18}$$

Considering $c_1 = c_2$, we have $h(c_1, X_1) = h(c_2, X_2)$.

If we are given $h(c_1, X_1) = h(c_2, X_2)$, we are able to prove the conditions mentioned in the theorem are necessary by showing contradictions occur when they are not satisfied. If $h(c_1, X_1) = h(c_2, X_2)$, we have:

$$h(c_1, X_1) - h(c_2, X_2) =$$
$$\sum_{x \in X_1} \left[\frac{f_{c_1 x} \cdot s_{c_1 x}}{\sum_{x \in X_1} f_{c_1 x} \cdot s_{c_1 x}}\right] \cdot g(x) - \sum_{x \in X_2} \left[\frac{f_{c_2 x} \cdot s_{c_2 x}}{\sum_{x \in X_2} f_{c_2 x} \cdot s_{c_2 x}}\right] \cdot g(x) = 0 \tag{19}$$

Firstly, assuming $S_1 \neq S_2$, for any $g(\cdot)$, we thereby have:

$$h(c_1, X_1) - h(c_2, X_2) = \sum_{x \in S_1 \cap S_2} \left[\frac{\exp\left(m_{c_1 x}\right) \sum_{y=x, y \in X_1} \exp\left(\mathbf{C}_{c_1 y}\right)}{\sum_{x \in S_1} \exp\left(m_{c_1 x}\right) \sum_{y=x, y \in X_1} \exp\left(\mathbf{C}_{c_1 y}\right)}\right.$$
$$\left. - \frac{\exp\left(m_{c_2 x}\right) \sum_{y=x, y \in X_2} \exp\left(\mathbf{C}_{c_2 y}\right)}{\sum_{x \in S_2} \exp\left(m_{c_2 x}\right) \sum_{y=x, y \in X_2} \exp\left(\mathbf{C}_{c_2 y}\right)}\right] \cdot g(x)$$
$$+ \sum_{x \in S_1 \setminus S_2} \left[\frac{\exp\left(m_{c_1 x}\right) \sum_{y=x, y \in X_1} \exp\left(\mathbf{C}_{c_1 y}\right)}{\sum_{x \in S_1} \exp\left(m_{c_1 x}\right) \sum_{y=x, y \in X_1} \exp\left(\mathbf{C}_{c_1 y}\right)}\right] \cdot g(x) \tag{20}$$
$$- \sum_{x \in S_2 \setminus S_1} \left[\frac{\exp\left(m_{c_2 x}\right) \sum_{y=x, y \in X_2} \exp\left(\mathbf{C}_{c_2 y}\right)}{\sum_{x \in S_2} \exp\left(m_{c_2 x}\right) \sum_{y=x, y \in X_2} \exp\left(\mathbf{C}_{c_2 y}\right)}\right] \cdot g(x) = 0$$

As Eq. (20) holds for any $g(\cdot)$, we may define another function $g'(\cdot)$ as shown in Eq. (5). If Eq. (20) holds, we also have:

$$h(c_1, X_1) - h(c_2, X_2) = \sum_{x \in S_1 \cap S_2} \left[\frac{\exp\left(m_{c_1 x}\right) \sum_{y=x, y \in X_1} \exp\left(\mathbf{C}_{c_1 y}\right)}{\sum_{x \in S_1} \exp\left(m_{c_1 x}\right) \sum_{y=x, y \in X_1} \exp\left(\mathbf{C}_{c_1 y}\right)}\right.$$
$$\left. - \frac{\exp\left(m_{c_2 x}\right) \sum_{y=x, y \in X_2} \exp\left(\mathbf{C}_{c_2 y}\right)}{\sum_{x \in S_2} \exp\left(m_{c_2 x}\right) \sum_{y=x, y \in X_2} \exp\left(\mathbf{C}_{c_2 y}\right)}\right] \cdot g'(x)$$
$$+ \sum_{x \in S_1 \setminus S_2} \left[\frac{\exp\left(m_{c_1 x}\right) \sum_{y=x, y \in X_1} \exp\left(\mathbf{C}_{c_1 y}\right)}{\sum_{x \in S_1} \exp\left(m_{c_1 x}\right) \sum_{y=x, y \in X_1} \exp\left(\mathbf{C}_{c_1 y}\right)}\right] \cdot g'(x)$$
$$- \sum_{x \in S_2 \setminus S_1} \left[\frac{\exp\left(m_{c_2 x}\right) \sum_{y=x, y \in X_2} \exp\left(\mathbf{C}_{c_2 y}\right)}{\sum_{x \in S_2} \exp\left(m_{c_2 x}\right) \sum_{y=x, y \in X_2} \exp\left(\mathbf{C}_{c_2 y}\right)}\right] \cdot g'(x)$$
$$= \sum_{x \in S_1 \cap S_2} \left[\frac{\exp\left(m_{c_1 x}\right) \sum_{y=x, y \in X_1} \exp\left(\mathbf{C}_{c_1 y}\right)}{\sum_{x \in S_1} \exp\left(m_{c_1 x}\right) \sum_{y=x, y \in X_1} \exp\left(\mathbf{C}_{c_1 y}\right)}\right. \tag{21}$$
$$\left. - \frac{\exp\left(m_{c_2 x}\right) \sum_{y=x, y \in X_2} \exp\left(\mathbf{C}_{c_2 y}\right)}{\sum_{x \in S_2} \exp\left(m_{c_2 x}\right) \sum_{y=x, y \in X_2} \exp\left(\mathbf{C}_{c_2 y}\right)}\right] \cdot g(x)$$
$$+ \sum_{x \in S_1 \setminus S_2} \left[\frac{\exp\left(m_{c_1 x}\right) \sum_{y=x, y \in X_1} \exp\left(\mathbf{C}_{c_1 y}\right)}{\sum_{x \in S_1} \exp\left(m_{c_1 x}\right) \sum_{y=x, y \in X_1} \exp\left(\mathbf{C}_{c_1 y}\right)}\right] \cdot [g(x) + 1]$$
$$- \sum_{x \in S_2 \setminus S_1} \left[\frac{\exp\left(m_{c_2 x}\right) \sum_{y=x, y \in X_2} \exp\left(\mathbf{C}_{c_2 y}\right)}{\sum_{x \in S_2} \exp\left(m_{c_2 x}\right) \sum_{y=x, y \in X_2} \exp\left(\mathbf{C}_{c_2 y}\right)}\right] \cdot [g(x) - 1] = 0$$

As Eq. (20) equals Eq. (21), we have:

$$\sum_{x \in S_1 \setminus S_2} \left[ \frac{\exp\left(m_{c_1 x}\right) \sum_{y=x, y \in X_1} \exp\left(\mathbf{C}_{c_1 y}\right)}{\sum_{x \in S_1} \exp\left(m_{c_1 x}\right) \sum_{y=x, y \in X_1} \exp\left(\mathbf{C}_{c_1 y}\right)} \right]$$
$$+ \sum_{x \in S_2 \setminus S_1} \left[ \frac{\exp\left(m_{c_2 x}\right) \sum_{y=x, y \in X_2} \exp\left(\mathbf{C}_{c_2 y}\right)}{\sum_{x \in S_2} \exp\left(m_{c_2 x}\right) \sum_{y=x, y \in X_2} \exp\left(\mathbf{C}_{c_2 y}\right)} \right] = 0 \tag{22}$$

Obviously, the above equation does not hold as softmax function is positive. Thus, $S_1 \neq S_2$ is not true. We may now assume $S_1 = S_2 = S$. Eliminating the irrational terms in Eq. (20), we have:

$$\sum_{x \in S_1 \cap S_2} \left[ \frac{\exp\left(m_{c_1 x}\right) \sum_{y=x, y \in X_1} \exp\left(\mathbf{C}_{c_1 y}\right)}{\sum_{x \in S_1} \exp\left(m_{c_1 x}\right) \sum_{y=x, y \in X_1} \exp\left(\mathbf{C}_{c_1 y}\right)} \right.$$
$$\left. - \frac{\exp\left(m_{c_2 x}\right) \sum_{y=x, y \in X_2} \exp\left(\mathbf{C}_{c_2 y}\right)}{\sum_{x \in S_2} \exp\left(m_{c_2 x}\right) \sum_{y=x, y \in X_2} \exp\left(\mathbf{C}_{c_2 y}\right)} \right] \cdot g(x) = 0 \tag{23}$$

Thus we know each term in the summation equals zero:

$$\frac{\exp\left(m_{c_1 x}\right) \sum_{y=x, y \in X_1} \exp\left(\mathbf{C}_{c_1 y}\right)}{\sum_{x \in S_1} \exp\left(m_{c_1 x}\right) \sum_{y=x, y \in X_1} \exp\left(\mathbf{C}_{c_1 y}\right)}$$
$$- \frac{\exp\left(m_{c_2 x}\right) \sum_{y=x, y \in X_2} \exp\left(\mathbf{C}_{c_2 y}\right)}{\sum_{x \in S_2} \exp\left(m_{c_2 x}\right) \sum_{y=x, y \in X_2} \exp\left(\mathbf{C}_{c_2 y}\right)} = 0 \tag{24}$$

Eq. (24) is equivalent to:

$$\frac{\sum_{y=x, y \in X_1} \exp\left(\mathbf{C}_{c_1 y}\right)}{\sum_{y=x, y \in X_2} \exp\left(\mathbf{C}_{c_2 y}\right)} = \frac{\exp\left(m_{c_2 x}\right) \sum_{x \in S_1} \exp\left(m_{c_1 x}\right) \sum_{y=x, y \in X_1} \exp\left(\mathbf{C}_{c_1 y}\right)}{\exp\left(m_{c_1 x}\right) \sum_{x \in S_2} \exp\left(m_{c_2 x}\right) \sum_{y=x, y \in X_2} \exp\left(\mathbf{C}_{c_2 y}\right)} \tag{25}$$

We may consider $S = \{s, s_0\}$ and assume $c_1 = s_0$, $c_2 = s$. We may also assume the feature similarity between central node and others as follows:

$$m_{c_1 x} = 1, \text{for } x \in S$$
$$m_{c_2 s} = 1, m_{c_2 s_0} = 2 \tag{26}$$

Consider $x = s$, we have:

$$\frac{\sum_{s \in X_1} \exp\left(\mathbf{C}_{c_1 s}\right)}{\sum_{s \in X_2} \exp\left(\mathbf{C}_{c_2 s}\right)} = \frac{e[e \sum_{s \in X_1} \exp\left(\mathbf{C}_{c_1 s}\right) + e \sum_{s_0 \in X_1} \exp\left(\mathbf{C}_{c_1 s_0}\right)]}{e[e \sum_{s \in X_2} \exp\left(\mathbf{C}_{c_2 s}\right) + e^2 \sum_{s_0 \in X_2} \exp\left(\mathbf{C}_{c_2 s_0}\right)]} \tag{27}$$

As the learning of $\mathbf{C}$ is independent of feature mapping, and the computation of attention coefficients, $\exp\left(\mathbf{C}_{cx}\right)$ can be any positive value. By setting the exponential values in the above equation as $a$, which is a positive value. We have $\frac{\mu_1(s)}{\mu_2(s)} = \frac{|X_1|}{|X_2| - n + ne}$. Similar with Eq. (12), $c_1 \neq c_2$ is not true. Since $c_1 = c_2 = c$, Eq. (25) can be rewritten as:

$$\frac{\sum_{y=x, y \in X_1} \exp\left(\mathbf{C}_{cy}\right)}{\sum_{y=x, y \in X_2} \exp\left(\mathbf{C}_{cy}\right)} = \frac{\sum_{x \in S_1} \exp\left(m_{cx}\right) \sum_{y=x, y \in X_1} \exp\left(\mathbf{C}_{cy}\right)}{\sum_{x \in S_2} \exp\left(m_{cx}\right) \sum_{y=x, y \in X_2} \exp\left(\mathbf{C}_{cy}\right)} = const > 0. \tag{28}$$

By setting $const$ as $\frac{1}{q}$ and $\exp\left(\mathbf{C}_{cy}\right) = \phi(\mathbf{C}_{cy})$, we have $q \sum_{y=x, y \in X_1} \phi(\mathbf{C}_{cy}) = \sum_{y=x, y \in X_2} \phi(\mathbf{C}_{cy})$. $\qquad \square$

## C  Proof of Corollary 1

According to Theorem 1, we denote $X_1 = (S, \mu_1)$, $X_2 = (S, \mu_2)$, $c \in S$. We also assume $\sum_{y=x, y \in X_1} f_{c_1 y} - \sum_{y=x, y \in X_2} f_{c_2 y} = q[\sum_{y=x, y \in X_2} s_{c_2 y} - \sum_{y=x, y \in X_1} s_{c_1 y}]$, for $q = \frac{r_s}{r_f}$. When $\mathcal{T}$ uses the weights obtained solely by Eq. (7) in the manuscript to aggregate node features, it is easy to verify $\sum_{x \in X_1} \alpha_{cx} f(x) = \sum_{x \in X_2} \alpha_{cx} f(x)$, i.e., $\mathcal{T}$ cannot distinguish the structures satisfied the aforementioned conditions. When $\mathcal{T}$ uses the Conjoint Attentions shown in Eq. (9) in the manuscript for feature aggregation and the corresponding attention scores are obtained by the *Implicit Strategy*

Table 1: Characteristics of the testing datasets used in our experiments

|  | Cora | Cite | Pubmed | CoauthorCS | OGB-Arxiv |
|---|---|---|---|---|---|
| $N$ | 2708 | 3327 | 19717 | 18333 | 169343 |
| $|E|$ | 5429 | 4732 | 44338 | 327576 | 1166243 |
| $D$ | 1433 | 3703 | 500 | 6805 | 128 |
| $C$ | 7 | 6 | 3 | 15 | 40 |
| **Training Nodes** | 140 | 120 | 60 | 300 | 90941 |
| **Validation Nodes** | 500 | 500 | 500 | 500 | 29799 |
| **Test Nodes** | 1000\2708 | 1000\3327 | 1000\19717 | 1000\18333 | 48603\169343 |

(Eq. (7) in the manuscript), we have $\sum_{x \in X_1} \alpha_{cx} f(x) - \sum_{x \in X_2} \alpha_{cx} f(x) = \epsilon(\frac{1}{|X_1|} - \frac{1}{|X_2|})\alpha_{cc} f(c)$, where $|X_1| = |\mathcal{N}_1|$, and $|X_2| = |\mathcal{N}_2|$. Since $|X_1| \neq |X_2|$, $\sum_{x \in X_1} \alpha_{cx} f(x) - \sum_{x \in X_2} \alpha_{cx} f(x) \neq 0$, meaning the aggregation function $\mathcal{T}$ that is based on Eqs. (7) and (9) in the manuscript, can successfully distinguish all the structures that cannot be discriminated by $\mathcal{T}$ solely based on Eq. (7) in the manuscript. Similarly, when the proposed Conjoint Attention (Eq. (9) in the manuscript) utilizing the *Explicit Strategy* (Eq. (8) in the manuscript), we can prove such aggregation function also can distinguish those distinct multisets that cannot be discriminated by the one solely based on the *Explicit Strategy*. □

## D   More details on the experiments

In this section, how the experiments used to validate the effectiveness of the proposed Graph conjoint attention networks (CATs) utilizing different Conjoint Attentions are set up is introduced with more details.

### D.1   Dataset description

Five network datasets, which are Cora, Cite, Pubmed [7, 11], CoauthorCS [12], and OGB-Arxiv [3], are used in our experiments to validate the effectiveness of different approaches. In Cora, Cite, Pubmed, and OGB-Arxiv, vertices, edges, and vertex features represent the documents, citations between pairwise documents, and the bag-of-words representations of the documents, respectively. While, in CoauthorCS, the authors, author-author collaborations, and the keywords of the collaborated papers are respectively represented as nodes, edges, and node features. The statistics of these benchmarking datasets are summarized in Table 1, where $N$, $|E|$, $D$, and $C$ denote the number of vertices, edges, vertex features, and the number of classes in each dataset, respectively.

### D.2   Learning tasks for evaluation

Two learning tasks, that are semi-supervised node classification and semi-supervised node clustering (community detection), are considered in our experiments to validate the effectiveness of different approaches. To set up the experiments, we closely follow the paradigms used in previous works [3, 13, 17]. All the datasets are split into three parts: training, validation, and testing. In both two learning tasks, we use the same split of training set in each dataset. For node classification tasks, a fraction of nodes are used for testing in each dataset, but all are considered in node clustering tasks. Specifically, in Cora, Cite, Pubmed, and CoauthorCS, only 20 labeled nodes per class are used for training, but all the feature vectors. As for testing, we use 1000 nodes in each dataset to build the testing split for node classification task. For dataset OGB-Arxiv, we use a practical split strategy provided in [3], segmenting the nodes which represent the academic papers, according to the publication years. Papers published up to 2017 are in the training split. While those published in 2018 and 2019 are used for validation and test for semi-supervised node classification, respectively.

The reason that we use the aforementioned two learning tasks in our experiments is they may test the effectiveness of different approaches in a complementary manner. Semi-supervised node classification may reflect the predictive accuracy of a learner from a local perspective, as a fraction of nodes in each dataset are used in the testing phase. In contrast, semi-supervised node clustering may indicate the overall learning performance of an approach, when a small number of node labels are used in the training stage. As all nodes are considered, semi-supervised node clustering may involve more

Table 2: Key settings of different approaches

| | Cora | Cite | Pubmed | CoauthorCS | OGB-Arxiv |
|---|---|---|---|---|---|
| **MoNet** | dropout=0.5
lr=0.01
hidden=16 | dropout=0.5
lr=0.01
hidden=16 | dropout=0.5
lr=0.01
hidden=16 | dropout=0.75
lr=0.005
hidden=32 | dropout=0.75
lr=0.005
hidden=256 |
| **GCN** | dropout=0.5
lr=0.01
hidden=32 | dropout=0.5
lr=0.01
hidden=32 | dropout=0.5
lr=0.01
hidden=32 | dropout=0.5
lr=0.005
hidden=32 | dropout=0.5
lr=0.005
hidden=256 |
| **GraphSAGE** | dropout=0.5
lr=0.01
hidden=16 | dropout=0.5
lr=0.01
hidden=16 | dropout=0.5
lr=0.01
hidden=16 | dropout=0.5
lr=0.01
hidden=32 | dropout=0.5
lr=0.01
hidden=256 |
| **JKNet** | dropout=0.5
lr=0.005
hidden=32 | dropout=0.5
lr=0.005
hidden=32 | dropout=0.5
lr=0.005
hidden=32 | dropout=0.5
lr=0.005
hidden=32 | dropout=0.5
lr=0.005
hidden=256 |
| **GAT and variants** | dropout=0.6
lr=0.005
hidden=8
#hidden heads=8
#output heads=1 | dropout=0.6
lr=0.005
hidden=8
#hidden heads=8
#output heads=1 | dropout=0.6
lr=0.005
hidden=8
#hidden heads=8
#output heads=1 | dropout=0.6
lr=0.005
hidden=32
#hidden heads=8
#output heads=1 | dropout=0.75
lr=0.002
hidden=256
#hidden heads=3
#output heads=3 |
| **APPNP** | dropout=0.6
lr=0.01
hidden=64 | dropout=0.6
lr=0.01
hidden=64 | dropout=0.6
lr=0.01
hidden=64 | dropout=0.6
lr=0.01
hidden=64 | dropout=0.75
lr=0.01
hidden=256 |
| **SGC** | lr=0.05 | lr=0.05 | lr=0.05 | lr=0.05 | lr=0.05 |
| **ARMA** | dropout=0.75
lr=0.01
hidden=64 | dropout=0.25
lr=0.01
hidden=64 | dropout=0.25
lr=0.01
hidden=64 | dropout=0.75
lr=0.01
hidden=64 | dropout=0.75
lr=0.01
hidden=256 |
| **GIN** | dropout=0.6
lr=0.01
hidden=64 | dropout=0.6
lr=0.01
hidden=64 | dropout=0.6
lr=0.01
hidden=64 | dropout=0.6
lr=0.01
hidden=64 | dropout=0.75
lr=0.01
hidden=256 |
| **CAT** | dropout=0.6
lr=0.01
hidden=8
#hidden heads=8
#output heads=1 | dropout=0.6
lr=0.01
hidden=8
#hidden heads=8
#output heads=1 | dropout=0.6
lr=0.01
hidden=8
#hidden heads=8
#output heads=1 | dropout=0.6
lr=0.01
hidden=8
#hidden heads=8
#output heads=1 | dropout=0.75
lr=0.002
hidden=256
#hidden heads=3
#output heads=3 |

potential structures in the graph, so that the testing stage is more challenging and the power of different graph learning methods can be comprehensively evaluated.

### D.3 Detailed settings of the graph neural networks

The proposed CATs utilizing different Conjoint Attentions have been compared with a number of prevalent baselines, including Arma filter GNN (ARMA) [1], Simplified graph convolutional Networks (SGC) [14], Personalized Pagerank GNN (APPNP) [6], Graph attention networks (GAT) [13], Jumping knowledge networks (JKNet) [16], Graph convolutional networks (GCN) [5], GraphSAGE [2], Mixture model CNN (MoNet) [8], and Graph isomorphism network (GIN) [15]. Besides, we also construct several variants of GAT which use the concatenation of original node features and structural embeddings [9, 10] to learn representations in all the testing datasets. To perform an unbiased comparison, we use the publicly released source codes to implement all the selected baselines. All the compared baselines use a two-layer network structure, i.e. the output layer followed by only one hidden layer, to learn representations for the downstream tasks. As for the tunable parameters of each baseline, we either use the default settings, or attempt to find the ones that lead the baseline to learn the best representations in each dataset. Here, we summarize the configurations of pivotal settings of all the compared baselines in Table 2, where lr and hidden stand for the learning rate and the dimension of hidden layer, respectively. As for the settings of CATs, we generally configure them as what we have done with GAT (see Table 2). Specifically, the maximum number of training epochs is set to 1500. The learning rate in OGB-Arxiv is 0.002, and that in the rest of the datasets is set to 0.01. All graph neural networks are initialized using Glorot initialization and trained to minimize cross-entropy on the training nodes using the Adam SGD optimizer [4]. And all the neural networks are trained on a single graphics card, NVIDIA RTX 3090 with 24GB, and are done in the following software environment: Python 3.8, PyTorch 1.8.1, and CUDA 11.1.

## E Supplementary results

### E.1 C pertaining to the relevance of input features

As mentioned in the manuscript, $\mathbf{C}$ can be any factors representing the node-node relevance that is not considered by the GNN. To further investigate whether the proposed graph neural network is effective using different interventions external to the GNN, we compute $\mathbf{C}$ based on the cosine

similarity (FS) between input features (i.e., $\mathbf{X}$) and then let different variants of CATs use it to perform semi-supervised node classification and clustering tasks in all the testing datasets. The performance comparisons between CATs and GATs are summarized in Tables 3 and 4. As the tables show, the performances on both classification and clustering are improved when CATs using similarity in terms of input features (CAT-I-FS and CAT-E-FS) are compared with GAT. However, they don't perform so well as GAT using concatenation of input feature and structural embeddings in some of the datasets. As the similarity of input features is similar to the correlation of layer-wise node features internal to the GNN, the performance improvement of CATs is not so evident as that of GAT using concatenation of input feature and structural embeddings. The obtained results also indicate that considering structural properties in attention-based GNNs may improve their learning performance.

Table 3: Performance comparison on semi-supervised node classification between GATs and CATs using input feature similarity.

|  | Cora | Cite | Pubmed | CoauthorCS | OGB-Arxiv |
|---|---|---|---|---|---|
| GAT | $83.84 \pm 0.61$ | $70.36 \pm 0.42$ | $81.50 \pm 0.47$ | $92.80 \pm 0.41$ | $72.39 \pm 0.07$ |
| GAT-$k$-Lap | $84.10 \pm 0.24$ | $71.18 \pm 0.52$ | $82.56 \pm 0.30$ | $92.70 \pm 0.31$ | $72.47 \pm 0.06$ |
| GAT-NetMF | $84.44 \pm 0.19$ | $70.94 \pm 0.16$ | $81.90 \pm 0.33$ | $93.16 \pm 0.27$ | $72.42 \pm 0.08$ |
| GAT-Deep | $83.68 \pm 0.67$ | $69.70 \pm 0.57$ | $80.13 \pm 0.26$ | $92.93 \pm 0.17$ | $72.79 \pm 0.09$ |
| CAT-I-FS | $84.86 \pm 0.30$ | $72.54 \pm 0.47$ | $82.79 \pm 0.34$ | $93.08 \pm 0.24$ | $72.57 \pm 0.02$ |
| CAT-E-FS | $84.15 \pm 0.23$ | $71.68 \pm 0.45$ | $82.33 \pm 0.27$ | $93.33 \pm 0.23$ | $72.52 \pm 0.03$ |

Table 4: Performance comparison on semi-supervised node clustering between GATs and CATs using input feature similarity.

|  | Cora | Cite | Pubmed | CoauthorCS | OGB-Arxiv |
|---|---|---|---|---|---|
| GAT | $81.39 \pm 0.18$ | $69.20 \pm 0.28$ | $80.88 \pm 0.33$ | $90.09 \pm 0.15$ | $76.04 \pm 0.38$ |
| GAT-$k$-Lap | $80.66 \pm 0.31$ | $69.56 \pm 0.34$ | $81.59 \pm 0.09$ | $89.83 \pm 0.18$ | $76.21 \pm 0.06$ |
| GAT-NetMF | $81.75 \pm 0.26$ | $68.96 \pm 0.21$ | $81.74 \pm 0.18$ | $89.85 \pm 0.21$ | $76.06 \pm 0.07$ |
| GAT-Deep | $81.08 \pm 0.41$ | $68.27 \pm 0.06$ | $80.55 \pm 0.11$ | $89.70 \pm 0.27$ | $76.91 \pm 0.15$ |
| CAT-I-FS | $81.83 \pm 0.23$ | $70.36 \pm 0.43$ | $81.89 \pm 0.37$ | $90.20 \pm 0.17$ | $76.33 \pm 0.03$ |
| CAT-E-FS | $81.50 \pm 0.26$ | $70.37 \pm 0.43$ | $81.91 \pm 0.34$ | $90.01 \pm 0.19$ | $76.18 \pm 0.33$ |

## E.2 Experiment on space complexity

When using the proposed CATs to learn representations from graph data, we allow the learning of $\mathbf{C}_{ij}$ (i.e., Eq. (2) or (3) in the manuscript) to be simultaneously done with the training of the neural network. To do so, the loss function of CATs can be simply reformulated as $L = L_{predict} + \lambda L_{ext}$, where $L_{predict}$ is the task-specific loss, $L_{ext}$ is the loss for learning all $\mathbf{C}_{ij}$, and $\lambda$ is the balancing parameter. To further reduce the computational burden in the training stage, we use a low-dimensional matrix $\mathbf{V}$ (with dimension $N$-by-$C$ in our experiments) to approximate $\mathbf{C}$ (see Eq. (2) or (3) in the manuscript). Aiming at investigating whether the proposed CATs utilizing the aforementioned training strategy can be used in massive graph datasets, we record the number of model parameters of

Table 5: Model comparison between GAT and CAT using different learning strategies

|  |  | Cora | Cite | Pubmed | CoauthorCS | OGB-Arxiv |
|---|---|---|---|---|---|---|
| **GAT** | # Parameters | 92430 | 237644 | 32454 | 1746974 | 384280 |
|  | Space consumption | 1.1GB | 1.2GB | 1.2GB | 2.3GB | 5.6GB |
| **CAT-I-SC** | # Parameters | 111285 | 257505 | 91504 | 2021484 | 7158006 |
|  | Space consumption | 0.8GB | 1.0GB | 1.3GB | 2.5GB | 21.4GB |
| **CAT-I-MF** | # Parameters | 111285 | 257505 | 91504 | 2021484 | 7158006 |
|  | Space consumption | 1.0GB | 1.1GB | 1.3GB | 2.5GB | 6.3GB |
| **CAT-E-SC** | # Parameters | 111267 | 257487 | 91486 | 2021466 | 7158002 |
|  | Space consumption | 1.1GB | 1.3GB | 1.4GB | 2.5GB | 21.4GB |
| **CAT-E-MF** | # Parameters | 111267 | 257487 | 91486 | 2021466 | 7158002 |
|  | Space consumption | 1.2GB | 1.3GB | 1.4GB | 2.5GB | 6.6GB |

CATs and their memory consumption in all the testing datasets and compare it with the closely related attention-based GNN, GAT. The corresponding results are summarized in Table 5. Compared with GAT, CATs use more learnable parameters to complete the task of representation learning in different datasets. Such growth of parameters in CATs is mainly because CATs have to simultaneously learn structural interventions (e.g., $\mathbf{C}_{ij}$ in Eq. (2) or (3) in the manuscript) for the computation of Conjoint Attentions. As for the space complexity, how much video memory is used by CATs jointly determined by the graph neural architecture and the learning of $\mathbf{C}_{ij}$. When a simple but effective paradigm, e.g., MF shown in Eq. (2) in the manuscript is used for learning $\mathbf{C}_{ij}$, the memory consumption of CATs is very close to that of GAT, although they have more parameters involved into the back-propagation process. For example, in OGB-Arxiv, which is the largest dataset used in our experiments, CATs take only 1GB more than GAT does to learn representations. Given the competitive results on space complexity, it is said that the proposed Conjoint Attentions and corresponding CATs can be used in massive graph data for effective representation learning.

### E.3 Sensitivity test on $\lambda$

As structural interventions ($\mathbf{C}$) are learnable, we allow the learning of $\mathbf{C}$ to be simultaneously done with the training of CATs. Thus, the loss function of CATs becomes $L = L_{predict} + \lambda L_{ext}$, where $L_{predict}$ is the task-specific loss (e.g., cross entropy loss for node classification), $L_{ext}$ can be the sum of all possible items shown in either Eq. (2) or (3), and $\lambda$ is a balancing parameter for controlling the relative significance of learning $\mathbf{C}_{ij}$. To investigate the model sensitivity against $\lambda$, we set $\lambda = [10^{-4}, 10^{-2}, 10^{-1}, 1, 5, 10, 20, 50, 100]$ and run CATs in all datasets to test their performance on different settings of $\lambda$. The results have been plotted in Fig. 1. As depicted, all the versions of CATs may perform robustly under a wide range settings of $\lambda$.

Obtaining the presented results might be due to the following reasons. In this paper, we mostly use semi-supervised learning tasks to test the effectiveness of different approaches. This means only a very low proportion of node labels (e.g., 140 out of 2708 in dataset Cora) are used to compute $L_{predict}$ and consequently only the gradients related to this small number of labeled nodes will be computed in the back propagation stage of CATs in each training epoch. Besides, a proportion of the gradients related to the labeled nodes might still be set to zero due to the dropout mechanism in each layer of the GNN. Given the mentioned property of learning tasks and the dropout mechanism used by GNN, in each training epoch, the gradients related to $L_{predict}$ contribute very limitedly to the learning of $\mathbf{C}_{ij}$. In contrast, $L_{ext}$ sums all the $\mathbf{C}_{ij}$s and we do not apply dropout on the training of $L_{ext}$, i.e., the learning of $\mathbf{C}_{ij}$. As a result, the learning of $\mathbf{C}_{ij}$ is dominantly determined by $L_{ext}$, i.e., Eqs. (2) or (3) in the manuscript, and the proposed CATs are less sensitive to the changes of $\lambda$. Changing the settings of $\lambda$ may only affect the convergence of learning $\mathbf{C}_{ij}$. For simplicity, we set $\lambda = 0.01$ for CATs in our experiments.

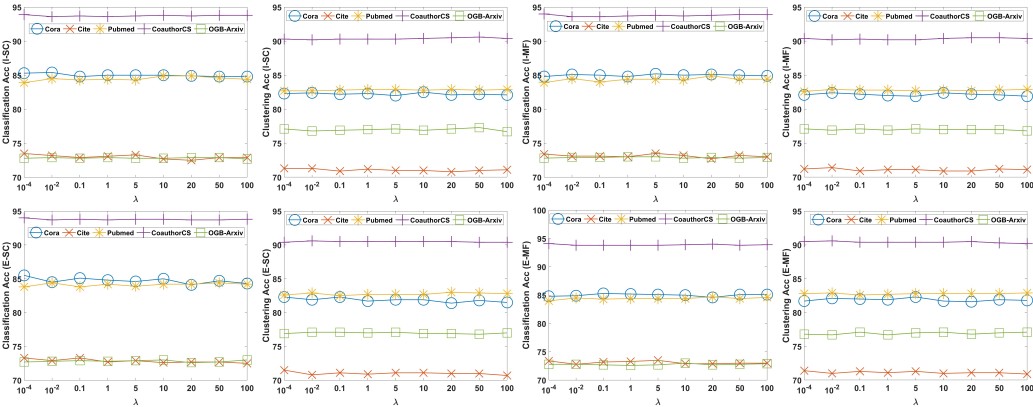

Figure 1: Sensitivity test on $\lambda$

# F Computational complexity

The computational complexity of the proposed CATs is mainly determined by the computation of attention layers and structural interventions. For the computation of each attention head in the graph neural network, its complexity for learning $D^{l+1}$ features for each node is same to the classical GAT, and can be represented as $O(N D^l D^{l+1} + (|E| + e) D^{l+1})$, where $e$ represents average degree of the nodes. If there are $K$ attention heads used, the complexity becomes $O(K(N D^l D^{l+1} + (|E| + e) D^{l+1}))$. As for the structural interventions for computing Conjoint Attentions, i.e., $\mathbf{C}_{ij}$, the complexity depends on its learning method. For example, when either matrix factorization or subspace learning method (Eq. (2) or (3) in the manuscript) is used, the complexity for learning each $\mathbf{C}_{ij}$ in each epoch is approximately $O(2(C + 1)N + C)$, where we assume the dimension of $\mathbf{V}$ that is used for approximating $\mathbf{C}$ is $N$-by-$C$.