# OpenReview forum: "Learning Conjoint Attentions for Graph Neural Nets"
_NeurIPS.cc/2021/Conference — NeurIPS 2021 Poster_

### Official Review · Reviewer_wBgV · 2021-07-08

**Rating:** 7
**Confidence:** 4

**Summary:**

This paper proposes a new technique for augmenting Graph Attentions with information other than node feature embedding in GNNs. The Conjoint Attentions (CAs) proposed in this paper parametrize the similarity hints between nodes, such as adjacency matrix, in the form of matrix factorization, and modify the attention weight \alpha calculated by ordinary graph attention.
This paper shows that CAs (with additional learnable parameters to the self-link terms) have theoretically higher expressive power than existing GNNs such as GAT.
Experimental results show that the proposed CAs have higher generalization performance than several existing GNNs on five representative datasets.


**Ethical Concerns:**

No specific concerns

**Limitations And Societal Impact:**

I felt that there was not much discussion about the limitation of the proposed method (though some are discussed in the appendix). Personally, I'm interested in the following questions:
- Is there any other typical and useful outside enlightenment other than the adjacency matrix?
- If there are multiple enlightenments, how should we weigh them? Which one should be selected?
- If the enlightenment is adversarial (e.g., false or noisy), how can we detect or mitigate the performance degradation?


**Main Review:**

## Originality:
Traditional graph Attentions are solely controlled by the similarity of node feature embedding. As far as I know, this paper is the first attempt to identify this problem: a unique and valuable problem proposal.
Also, the idea of adding a new control term to the existing attention in the form of Eqs.(7-9) is new, at least in my knowledge of GNN research. I find it interesting

However, it is a pity that only the adjacency matrix is studied as enlightenment. ``A'' could be said to be information inside the graph data, depending on how a reader looks at it. I think the paper would be strengthened more if there were examples of information that is clearly outside the graph data.

## Quality:
I could not find any technically obvious mistakes in this paper. However, I need to confess I cannot follow the proofs of theorems.

Many related studies often include Graph Isomorphism Network (GIN[33]) as one of the most expressive GNNs for comparison. In my experience, GIN does not necessarily have the numerically highest generalization performance, but additional comparisons are definitely desirable to support the high expressive power shown in Section 4.
Notice: I believe that the value of this paper would not change significantly even if CATs do not perform better than GIN.

## Clarity:
I could not understand the main ideas of this paper on the first reading.
One reason is that (i)  concrete examples of ``other possible enlightenment'' (first appear in introduction L32) and (ii) explanation of how they help attention are not explained even roughly until section 3 (L115, 147).
Ideally, I would like the introduction to be augmented so that the readers can have some intuitions about the outline of the proposed method at the introduction stage (I don't mean that it should be shown in equations explicitly). It is difficult for me to understand the intention of the equations at the first sight unless some intuitions are provided beforehand.

In Figure 3, I'm happy to see that the result is robust to hyperparameter \lambda choices.
But I think this figure can be moved to an appendix. As a reader, I can read as much information in the text of L308-312 alone as in figure 3. I would like to see additional experiments and/or discussion to limitation, future works in the main manuscript instead.

## Significance:
The experimental results show high test generalization performance of the proposed CAs, as expected from the theoretical conclusions, and I think these results support the claims of the paper well.
As mentioned in the Originality section, I think the paper would be more significant if there are additional examples that incorporate information from outside the graph. However, I think it is an important finding that just using the adjacency matrix to control attention in this way can improve GNN performance.

Section 4 proves that the expressive power is stronger than GAT, etc... This increases the significance of the paper.

The formulation of the proposed CA is simple enough to be fully understood when looking at Eqs. (5-9), which I think adds to the value of this paper.

## after feedbacks
I found the author feedback informative, yielding better understanding. I keep the original positive score.

**Time Spent Reviewing:**

5

---

> ### Author Response · Authors · 2021-08-09
> **Response to the comments from Reviewer wBgV - Part 1**
>
> $Q1$. However, it is a pity that only the adjacency matrix is studied as enlightenment. ``A'' could be said to be information inside the graph data, depending on how a reader looks at it. I think the paper would be strengthened more if there were examples of information that is clearly outside the graph data.
>
> $Response$: It is worth noting that the adjacency matrix is indeed an important part of graph data, but has yet to be properly exploited in empirical attention-based GNNs. Thus, we use $\mathbf C$ to learn node-node correlations pertaining to node cluster embeddings and self-representation coefficients (Eqns. (2) and (3)), and then have CATs use $\mathbf C$ to compute the conjoint attentions. As $\mathbf C$ has to be learned, we can partially treat such information as outside of the graph, as well as the GNN. In this manuscript, we refer such information that can be learned outside of GNNs and can then be used to compute conjoint attentions as external factors. In fact, many other types of factors that are not explicitly carried by the graph can be obtained by means of simple calculations, e.g., node-node similarity in terms of input features, first-order, or higher-order Markov matrix based on adjacency matrix, and even the average node-node similarity in terms of both node adjacency and input features. As long as such factors may effectively determine the node-node correlations differently from node embeddings in the GNN, it can be used by CATs and possibly obtain an improved learning performance. We will include such discussions into the final version of the paper.
>
> To further demonstrate the effectiveness of the proposed method, we let CATs learn node representations using node-node similarity with respect to input features (CAT-I-FS and CAT-E-FS), which is another possible form of external factors. The corresponding results are briefly shown in Table 1. As the table depicts, the proposed CATs outperform GAT in most of the datasets. The results reported here will be included into the final version of the paper.
>
> Table 1. Performance comparion between GAT and CATs using similarity of node feature
>
> |---------------------|--------------|--------------|--------------|--------------|----------------|
>
> | Node Classification | Cora         | Cite         | Pubmed       | CoauthorCS   | OGB-Arxiv      |
>
> |---------------------|--------------|--------------|--------------|--------------|----------------|
>
> | GAT                 | 83.84 (0.61) | 70.36 (0.42) | 81.50 (0.47) | 92.80 (0.41) | 72.39 (0.07) |
>
> | CAT-I-FS            | 84.86 (0.30) | 72.54 0.47)  | 82.79 (0.34) | 93.08 (0.24) | 72.57 (0.02) |
>
> | CAT-E-FS            | 84.15 (0.23) | 71.68 (0.45) | 82.33 (0.27) | 93.33 (0.23) | 72.52 (0.03) |
>
> |---------------------|--------------|--------------|--------------|--------------|----------------|
>
> | Node Clustering     | Cora         | Cite         | Pubmed       | CoauthorCS   | OGB-Arxiv |
>
> |---------------------|--------------|--------------|--------------|--------------|----------------|
>
> | GAT                 | 81.39 (0.18) | 69.20 (0.28) | 80.88 (0.33) | 90.09 (0.15) | 76.04 (0.38) |
>
> | CAT-I-FS            | 81.83 (0.23) | 70.36 (0.43) | 81.89 (0.37) | 90.20 (0.17) | 76.33 (0.03) |
>
> | CAT-E-FS            | 81.50 (0.26) | 70.37 (0.43) | 81.91 (0.34) | 90.01 (0.19) | 76.18 (0.33) |
>
> |---------------------|--------------|--------------|--------------|--------------|----------------|
>
> $Q2$. Many related studies often include Graph Isomorphism Network (GIN[33]) as one of the most expressive GNNs for comparison. In my experience, GIN does not necessarily have the numerically highest generalization performance, but additional comparisons are definitely desirable to support the high expressive power shown in Section 4. Notice: I believe that the value of this paper would not change significantly even if CATs do not perform better than GIN.
>
> $Response$: Thank you for the suggestion. As suggested by the reviewer, we have compared the proposed CATs with GIN and the corresponding results are summarized in the Table 2. As the table shows, CATs can outperform GIN in most datasets. We will include these results into the final version of the paper.
>
> Table 2. Performance comparion between GIN and CATs
>
> |-----------------------|------------------------|-----------------------|-----------------------|------------------------|-----------------------|
>
> |  Node Classification  |     Cora               |     Cite              |     Pubmed            |     CoauthorCS         |     OGB-Arxiv         |
>
> |-----------------------|------------------------|-----------------------|-----------------------|------------------------|-----------------------|
>
>
> |     GIN               |     81.58   (0.62)     |     66.90   (0.16)    |     80.76   (0.33)    |     93.03   (0.74)     |     64.02   (0.18)    |
>
>
> |     CAT-I-MF          |     85.38   (0.16)     |     73.22   (0.19)    |     83.90   (0.24)    |     93.74   (0.14)     |     72.89   (0.06)    |
>
>
> |     CAT-I-SC          |     85.50   (0.22)     |     73.18   (0.22)    |     84.28   (0.20)    |     93.70   (0.11)     |     72.85   (0.04)    |
>
>
> |     CAT-E-MF          |     85.56   (0.19)     |     73.24   (0.21)    |     83.60   (0.17)    |       93.40  (0.12)    |     72.81   (0.09)    |
>
>
> |     CAT-E-SC          |     85.40   (0.36)     |     73.02   (0.24)    |     84.02   (0.24)    |     93.30   (0.11)     |     72.83   (0.11)    |
>
> |-----------------------|------------------------|-----------------------|-----------------------|------------------------|-----------------------|
>
> |   Node Clustering     |     Cora               |     Cite              |     Pubmed            |     CoauthorCS         |     OGB-Arxiv         |
>
> |-----------------------|------------------------|-----------------------|-----------------------|------------------------|-----------------------|
>
> |     GIN               |     78.25   (0.46)     |     67.83   (0.15)    |     79.31   (0.35)    |     89.97   (0.26)     |     63.85   (0.18)    |
>
>
> |     CAT-I-MF          |       82.17  (0.11)    |     71.15   (0.12)    |     82.77   (0.07)    |     90.26   (0.22)     |     77.72   (0.07)    |
>
>
> |     CAT-I-SC          |     82.26   (0.13)     |     71.17   (0.15)    |     82.86   (0.07)    |     90.29   (0.21)     |     77.01   (0.16)    |
>
>
> |     CAT-E-MF          |     81.98   (0.19)     |     71.21   (0.12)    |     82.40   (0.08)    |     89.66   (0.22)     |     76.93   (0.16)    |
>
>
> |     CAT-E-SC          |     82.01   (0.24)     |     71.11   (0.24)    |     82.61 (0.14)      |     89.72   (0.15)     |     76.98   (0.08)    |
>
> |-----------------------|------------------------|-----------------------|-----------------------|------------------------|-----------------------|
>
> $Q3$. I could not understand the main ideas of this paper on the first reading. One reason is that (i) concrete examples of ``other possible enlightenment'' (first appear in introduction L32) and (ii) explanation of how they help attention are not explained even roughly until section 3 (L115, 147). Ideally, I would like the introduction to be augmented so that the readers can have some intuitions about the outline of the proposed method at the introduction stage (I don't mean that it should be shown in equations explicitly). It is difficult for me to understand the intention of the equations at the first sight unless some intuitions are provided beforehand.
>
> $Response$: Thank you for the suggestion. The term “enlightenment” introduced in this paper originally aims at offering a concrete representation of factors that can be acquired outside of the GNN and can then be used to compute conjoint attention coefficients. Nonetheless, we noted that the term “enlightenment” may lead to possible misunderstanding on the core idea of our proposed work in the manuscript. Thus, following the reviewer’s suggestion, we will not introduce “enlightenment” as a new term in the final version of the paper. Instead, in the revised manuscript, we will use commonly used term to introduce more factors with examples that can be potentially used for the computation of conjoint attentions. For example, node-node correlations pertaining to node cluster embeddings learned from adjacency matrix, higher order Markov matrix, and node-node similarity regarding to input features are all possible ones that can be used for computing conjoint attention coefficients. As such types of factors represent meaningful node-node relevance that is not determined by node embeddings, CATs are able to utilize them to compute novel attention scores (i.e., conjoint attentions) so as to aggregate the embeddings of neighbors that are heterogeneously related to the central node. As various relevance is considered by the proposed conjoint attention mechanisms, CATs may learn meaningful node representations and consequently perform better than GATs. Following the suggestion of the reviewer, we will introduce more examples of such factors and intuitively introduce how they may improve the learning performance of CATs in the section of Introduction.

---

> > ### Author Response · Authors · 2021-08-09
> > **Response to the comments from Reviewer wBgV - Part 2**
> >
> > $Q4$. In Figure 3, I'm happy to see that the result is robust to hyperparameter \lambda choices. But I think this figure can be moved to an appendix. As a reader, I can read as much information in the text of L308-312 alone as in figure 3. I would like to see additional experiments and/or discussion to limitation, future works in the main manuscript instead.
> >
> > $Response$: As suggested, we have compared CATs with GIN and GATs using the concatenation of node features (i.e., $\mathbf X$) and structural information. Besides, we have also investigated how CATs perform when using the similarity in terms of input node features to compute conjoint attentions. All the corresponding results have been uploaded to openreview (Table 1 and Table 3) and will be reported in the final version of the manuscript. As the tables show, the proposed CATs may perform robustly despite various external factors are used for computing conjoint attentions. Besides, we will discuss the limitations of CATs and more future works, which have been done in the appendix, instead of presenting Fig. 3 in the final version of main paper.
> >
> > Table 3. Performance comparion between GAT, GAT concatenating node feature and structural embeddings, and CATs
> >
> > |--------------------|-----------------|-----------------|-----------------|-----------------|-----------------------|
> >
> > | Node Classification|   Cora          |   Cite          |   Pubmed        |   CoauthorCS    |   OGB-Arxiv           |
> >
> > |--------------------|-----------------|-----------------|-----------------|-----------------|-----------------------|
> >
> > |   GAT              |   83.84 (0.61)  |   70.36 (0.42)  |   81.50 (0.47)  |   92.80 (0.41)  |   72.39 (0.07)  |
> >
> > |   GAT-k-Laplacian        |   84.1 (0.24)   |   71.18 (0.52)  |   82.56 (0.30)  |   92.70 (0.31)  |   72.47 (0.06)  |
> >
> > |   GAT-NetMF        |   84.44 (0.19)  |   70.94 (0.16)  |   81.90 (0.33)  |   93.16 (0.27)  |   72.42 (0.08)  |
> >
> > |   GAT-Deepwalk         |   83.68 (0.67)  |   69.70 (0.57)  |   80.13 (0.26)  |   92.93 (0.17)  |   72.79 (0.09)  |
> >
> > |   CAT-I-MF         |   85.38 (0.16)  |   73.22 (0.19)  |   83.90 (0.24)  |   93.74 (0.14)  |   72.89 (0.06)  |
> >
> > |   CAT-I-SC         |   85.50 (0.22)  |   73.18 (0.22)  |   84.28 (0.20)  |   93.70 (0.11)  |   72.85 (0.04)  |
> >
> > |   CAT-E-MF         |   85.56 (0.19)  |   73.24 (0.21)  |   83.60 (0.17)  |   93.40 (0.12)  |   72.81 (0.09)  |
> >
> > |   CAT-E-SC         |   85.40 (0.36)  |   73.02 (0.24)  |   84.02 (0.24)  |   93.30 (0.11)  |   72.83 (0.11)  |
> >
> > |--------------------|-----------------|-----------------|-----------------|-----------------|-----------------------|
> >
> > |   Node Clustering  |   Cora          |   Cite          |   Pubmed        |   CoauthorCS    |   OGB-Arxiv           |
> >
> > |--------------------|-----------------|-----------------|-----------------|-----------------|-----------------------|
> >
> > |   GAT              |   81.39 (0.18)  |   69.20 (0.28)  |   80.88 (0.33)  |   90.09 (0.15)  |   76.04 (0.38)  |
> >
> > |   GAT-k-Lap        |   80.66 (0.31)  |   69.56 (0.34)  |   81.59 (0.09)  |   89.83 (0.18)  |   76.21 (0.06)  |
> >
> > |   GAT-NetMF        |   81.75 (0.26)  |   68.96 (0.21)  |   81.74 (0.18)  |   89.85 (0.21)  |   76.06 (0.07)  |
> >
> > |   GAT-Deep         |   81.08 (0.41)  |   68.27 (0.06)  |   80.55 (0.11)  |   89.70 (0.27)  |   76.91 (0.15)  |
> >
> > |   CAT-I-MF         |   82.17 (0.11)  |   71.15 (0.12)  |   82.77 (0.07)  |   90.26 (0.22)  |   77.72 (0.07)  |
> >
> > |   CAT-I-SC         |   82.26 (0.13)  |   71.17 (0.15)  |   82.86 (0.07)  |   90.29 (0.21)  |   77.01 (0.16)  |
> >
> > |   CAT-E-MF         |   81.98 (0.19)  |   71.21 (0.12)  |   82.40 (0.08)  |   89.66 (0.22)  |   76.93 (0.16)  |
> >
> > |   CAT-E-SC         |   82.01 (0.24)  |   71.11 (0.24)  |   82.61 (0.14)  |   89.72 (0.15)  |   76.98 (0.08)  |
> >
> > |--------------------|-----------------|-----------------|-----------------|-----------------|-----------------------|
> >
> > $Q5$. The experimental results show high test generalization performance of the proposed CAs, as expected from the theoretical conclusions, and I think these results support the claims of the paper well. As mentioned in the Originality section, I think the paper would be more significant if there are additional examples that incorporate information from outside the graph. However, I think it is an important finding that just using the adjacency matrix to control attention in this way can improve GNN performance.
> >
> > $Response$: As suggested, CATs utilizing more external factors, e.g., the similarity in terms of input node features have been tested and the corresponding results have been presented in openreview (see Table 1) and will be reported in final version of the paper. Besides, many other types of factors, for example, first-order, or higher-order Markov matrix based on adjacency matrix, and the average node-node similarity in terms of both node adjacency and input features can be potentially incorporated with CATs. In the final version of the paper, we will introduce these possible factors together with examples and discuss their feasibility to compute conjoint attention coefficients for learning node representations.
> >
> > $Q6$. Is there any other typical and useful outside enlightenment other than the adjacency matrix?
> >
> > $Response$: As suggested by the reviewers, in the revised manuscript, will not introduce “enlightenment” as a new term to avoid any potential confusions. Instead, we refer those parameters that can be acquired outside of the GNN and can then be used to compute conjoint attentions as external factors. We sincerely apologize for that.
> >
> > As for the raised question, yes, many types of external factors (enlightenment), e.g., node-node similarity in terms of input features, first-order, or higher-order Markov matrix based on adjacency matrix, and the diffusion matrix, can be used to compute the conjoint attention scores. To better show the effectiveness of the proposed CATs, CATs utilizing more external factors, e.g., the similarity in terms of input node features have been tested and the corresponding results have been presented in openreview (see Table 1) and will be reported in final version of the paper.

---

> > > ### Author Response · Authors · 2021-08-09
> > > **Response to the comments from Reviewer wBgV - Part 3**
> > >
> > > $Q7$. If there are multiple enlightenments, how should we weigh them? Which one should be selected?
> > >
> > > $Response$: If multiple external factors (enlightenments) are given, we still have some effective methods for using them in the conjoint attention mechanisms. Multiple factors mean we have multiple data matrices. One possible way is to use a multi-view learning module [1] to learn a common relevance that is shared by the multiple data matrices. Taking Eqn. (2) as an example, the generic loss function for learning relevance from multiple data matrices can be defined as: $\mathbf C_{ij} = argmin_{\mathbf {VV}^T_{ij}} \sum_{d} \omega^d (\mathbf Y^d_{ij} - \mathbf U^d \mathbf V^T_{ij})^2$, where $\omega^d$ represents the relative significance of data matrix $\mathbf Y^d$, $\mathbf U^d$ is a low dimensional matrix helping $\mathbf V$ to approximate $\mathbf Y^d$, and $\omega^d$ can be predefined before the training or automatically learned alongside the learning of $\mathbf C_{ij}$. Another potential way is to learn $\mathbf C_{ij}$ individually for each data matrix outside of the GNN, and directly use Eq. (7) to compute the conjoint attention scores. The conjoint attention mechanism will automatically determine the relative significance between different factors. It should be noted there are many ways that can deal with the learning from multiple data matrices simultaneously automatic weighing multiple data matrices. And potentially they can be integrated with the proposed CATs. In our future works, we will attempt to investigate the performance when CATs are given multiple external factors that can be used to compute conjoint attentions and theoretically analyze how they may affect the capacity of CATs. And we will include the above discussions into the final version of the paper.
> > >
> > > $Q8$. If the enlightenment is adversarial (e.g., false or noisy), how can we detect or mitigate the performance degradation?
> > >
> > > $Response$: It is believed that the performance of CATs will be degenerated if the given external factor is false or noisy. And this might be one of the limitations of the proposed conjoint attention networks. However, some potential methods may mitigate the side-effect brought by possible false/noisy external factors. One is to utilize adversarial learning modules to alleviate the model vulnerability to external factor contamination. The other method is to consider learning effective factors from multiple data sources (i.e., multi-view). In previous works [1], multi-view learning has shown to be effective even when some view of data is contaminated. We will discuss this limitation raised by the reviewer in final version of the paper and attempt to improve the performance of CATs when CATs are confronted with contaminated factors as the future works.
> > >
> > > [1] C. Xu, D. Tao, and C. Xu, “A survey on multi-view learning,” arXiv:1304.5634, 2013.

---

> > > > ### Comment · Reviewer_wBgV · 2021-08-22
> > > > **Thank you for the clarification**
> > > >
> > > > Thank you authors for your very detailed comments.
> > > > These comments are useful and informative, a kind of eye-opening to me.
> > > >
> > > > I will keep my positive score.

---

> > > > > ### Author Response · Authors · 2021-08-23
> > > > > **Sincere gratitude for the detailed review**
> > > > >
> > > > > Thank you very much again for the detailed review and constructive comments. All the discussions and experiments that are suggested by the reviewers will be included in the revised version of the paper.

---

### Official Review · Reviewer_dyaY · 2021-07-12

**Rating:** 7
**Confidence:** 3

**Summary:**

This paper presents an extension of Graph Attention Networks (GAT) termed Conjoint Attention Networks (CAT). The attention mechanism presented in this paper combines the attention mechanism from GAT, computed from the node embeddings, with an extra attention mechanism that captures structural information too.

**Ethical Concerns:**

This paper does not present ethical issues.


**Limitations And Societal Impact:**

This paper doesn’t seem to have any obvious societal impact. The authors address the limitations of their method in the Appendix.

**Main Review:**

Strengths
+ The authors perform an experimental evaluation on a variety of graph datasets for two different tasks.
+ The authors do a theoretical analysis of their proposed method.

Weaknesses
- The main weakness of the paper is that it is not clear what is the motivation for CATs, and why the different methods for computing $\mathbf{C}_{ij}$ are chosen:

  - First of all, the authors mention that their attention mechanism can leverage “factors either inside or outside of GNNs” (Abstract, line 5). It isn’t clear what that means. What are factors inside the GNN? I assume deep node embeddings? What are the factors outside the GNN?

  - In the introduction (line 36), the authors mention that “previous graph attention mechanisms are incapable of learning effective attention scores when heterogeneous relevance is given”. What is “heterogeneous relevance”? It isn’t explained in the paper. Why are other attention mechanisms unable to learn effective attention scores? Why are CATs better in this aspect?

  - Again, in the introduction (line 41) the authors say that CATs are flexible to compute attention scores not only relying on the layer node features, but also leveraging “information possibly outside of the neural networks”. What is that information?

  - Section 3 should explain clearly what are CATs and their motivation. While the formulation of CATs is well explained, the choice of the conjoint attention scores $\mathbf{C}_{ij}$ and its motivation is unclear. In section 3.1 the authors explain that conjoint attentions can capture “enlightenment outside of GNNs”. Again, it isn’t clear what that means. The authors use the word “enlightenment” to refer to the information that the proposed attention mechanism can use, but it isn’t clear what that means.

  - In Equations (2) and (3) it seems that $C_{ij}$ is capturing structural information from the adjacency matrix of the graph, but that isn’t explained.

  - In the experiments the authors evaluate the MF and SC variants of their method, introduced in Equations (2) and (3), but they don’t explain much about their motivation. What is the difference between them? They seem to get similar experimental results, why?


Additional Questions:
* In Section 5.4 the authors say that the “external enlightenment C” is learnable, and that it can be simultaneously learned during the training of CATs. But why is it learnable? Does it have any learning parameters? Why can’t it be computed before training?
* In Figure 3, it appears that the value of $\lambda$ doesn’t have an effect, why is that the case? It isn’t explained. What should be the behaviour of $\mathbf{C}_{ij}$ as $\lambda$ increases (or decreases)?
* What are the different attention factors compared in Figure 2? What is the difference between SC and I-SC?

To summarize, the method seems to perform well, and the authors do a good evaluation in different datasets, but it is not clear why this method should work better than GATs. The authors keep mentioning "enlightenment", a concept that is not explained. The clarity of the paper and the quality of the explanations should improve in order to be accepted.


### Post Rebuttal:
The authors have answered my questions and have agreed to improve the clarity of the paper, which was my main concern. Because of that, I raise my score to a 7, assuming that the authors will improve the clarity of the paper in its final version.


**Time Spent Reviewing:**

4

---

> ### Author Response · Authors · 2021-08-09
> **Response to the comments from Reviewer dyaY - Part 1**
>
> $Q1$. First of all, the authors mention that their attention mechanism can leverage “factors either inside or outside of GNNs” (Abstract, line 5). It isn’t clear what that means. What are factors inside the GNN? I assume deep node embeddings? What are the factors outside the GNN?
>
> $Response$: Thank you very much for the suggestions. The mentioned content will be revised to avoid any potential confusions caused. Specifically, factors inside the GNN (internal factors) mainly refer to node embeddings in each layer of the GNN. Empirical attention-based GNNs (e.g., GAT) make use of node embeddings in the same layer to compute attention coefficients. In contrast, we regard those parameters that can be acquired outside of the GNN and can be used to compute conjoint attentions as external factors, for example, $\mathbf C_{ij}$ generated by $\mathbf V$ that is learned from adjacency matrix using matrix factorization (Eq. (2)), and self-representation coefficients learned from adjacency matrix using self-expressiveness modules (Eq. (3)). We propose to make both internal and external factors to learn the conjoint attentions used by the graph neural network. To avoid any potential confusions, in the revised manuscript, we will use more commonly-used terms to illustrate what is currently used by CATs to compute conjoint attentions, i.e., node-node correlations generated by node cluster embeddings (Eqn. (2)) [1], self-representation coefficients (Eqn. (3)) [2], and similarities in terms of input node features, when it needs.
>
> $Q2$. In the introduction (line 36), the authors mention that “previous graph attention mechanisms are incapable of learning effective attention scores when heterogeneous relevance is given”. What is “heterogeneous relevance”? It isn’t explained in the paper. Why are other attention mechanisms unable to learn effective attention scores? Why are CATs better in this aspect?
>
> $Response$: Thank you very much for the suggestion and awe apologize for the potential possible confusions caused. As suggested, we will explain the term “heterogeneous relevance” more clearly in the final version of the paper to avoid any potential confusions. It is known that empirical attention-based GNNs, for example, in GAT, compute attention scores based on the relevance between node embeddings. While, other forms of relevance, for example, the node self-representation coefficients that are learned by Eqn. (3) in the manuscript have been overlooked and underexplored by previous works. Thus, heterogeneous relevance in this paper was introduced to represent the node-node relevance brought by node embeddings inside the GNN, and other external factors, e.g., correlations generated by node cluster embeddings, and node self-representation coefficients that are learned by Eqns. (2) and (3).
>
> Intuitively, the mentioned relevance describes the node-node relationships from various aspects and can hence enable the GNN to learn node representations from neighbors that share similarities (of diverse forms) rather than limiting to layer-wise node embeddings solely. Taking the correlations pertaining to cluster structure (Eq. (2) in the manuscript) as an example, it is a meaningful latent structure hidden in the graph and those nodes belonging to the same cluster would potentially share higher cohesiveness. If cluster structure is considered, attention-based GNN is able to learn node representations that tend towards using embeddings of those neighbors in the same cluster. Using a practical example, such node representations can be learned from those articles whose topics are correlated and that are frequently cited by other similar articles in the citation networks, or from those social network users who frequently interact with each other in the same social group as well as sharing correlated hobbies in social graphs. Though such external factors might be beneficial to representation learning, previous graph attention networks cannot integrate them with ease together with the node embeddings for computing the attention coefficients. We therefore propose the conjoint attention (CA) mechanisms. Paying attention to similar embeddings of neighbors, the proposed conjoint attentions can also pay attention to those neighbors that share other types of relevance with the central node, when computing the attention coefficients. Thus meaningful node representations are expected to learn from those nodes that are heterogeneously relevant. In Section 4 of the manuscript, we theoretically prove that the expressive power of the proposed conjoint attentions is stronger than empirical graph attentions. And the experimental results also demonstrate to confirm that the proposed CATs perform better than the representative attention-based GNN, i.e., GAT.
>
> To further demonstrate the effectiveness of the proposed method, we let CATs learn node representations using node-node similarity with respect to input features (CAT-I-FS and CAT-E-FS), which is another possible form of relevance. The corresponding results are briefly summarized in the Table below. As the table depicts, the proposed CATs outperform GAT in most of the datasets. The results reported here will be included into the final version of the paper.
>
> |---------------------|--------------|--------------|--------------|--------------|----------------|
>
> | Node Classification | Cora         | Cite         | Pubmed       | CoauthorCS   | OGB-Arxiv      |
>
> |---------------------|--------------|--------------|--------------|--------------|----------------|
>
> | GAT                 | 83.84 (0.61) | 70.36 (0.42) | 81.50 (0.47) | 92.80 (0.41) | 72.39 (0.07) |
>
> | CAT-I-FS            | 84.86 (0.30) | 72.54 0.47)  | 82.79 (0.34) | 93.08 (0.24) | 72.57 (0.02) |
>
> | CAT-E-FS            | 84.15 (0.23) | 71.68 (0.45) | 82.33 (0.27) | 93.33 (0.23) | 72.52 (0.03) |
>
> |---------------------|--------------|--------------|--------------|--------------|----------------|
>
> | Node Clustering     | Cora         | Cite         | Pubmed       | CoauthorCS   | OGB-Arxiv |
>
> |---------------------|--------------|--------------|--------------|--------------|----------------|
>
> | GAT                 | 81.39 (0.18) | 69.20 (0.28) | 80.88 (0.33) | 90.09 (0.15) | 76.04 (0.38) |
>
> | CAT-I-FS          | 81.83 (0.23) | 70.36 (0.43) | 81.89 (0.37) | 90.20 (0.17) | 76.33 (0.03) |
>
> | CAT-E-FS         | 81.50 (0.26) | 70.37 (0.43) | 81.91 (0.34) | 90.01 (0.19) | 76.18 (0.33) |
>
> |---------------------|--------------|--------------|--------------|--------------|----------------|
>
> $Q3$. Again, in the introduction (line 41) the authors say that CATs are flexible to compute attention scores not only relying on the layer node features, but also leveraging “information possibly outside of the neural networks”. What is that information?
>
> $Response$: Thank you very much for the suggestion and we apologize for the potential possible confusions caused. In the final version of the paper, we will explain what information outside of the neural networks can be utilized by CATs with more examples, so as to avoid any potential confusions. Specifically, the information outside of the neural networks contains the external factors that can be acquired outside of the GNN, examples include node-node correlations generated by node cluster embeddings, and node self-representation coefficients learned from adjacency matrix (Eqns. (2) and (3)). In addition, we let CATs learn node representations using node-node similarity with respect to input features, which differs from the structural information mentioned above. And the corresponding results have been presented in the table above.

---

> > ### Author Response · Authors · 2021-08-09
> > **Response to the comments from Reviewer dyaY - Part 2**
> >
> > $Q4$. Section 3 should explain clearly what are CATs and their motivation. While the formulation of CATs is well explained, the choice of the conjoint attention scores and its motivation is unclear.
> >
> > $Response$: Thank you very much for pointing this out and we apologize for any potential confusions caused. In Section 3 of the revised manuscript, we will clearly illustrate the motivation of conjoint attentions and the intuition to the choice of different conjoint attention mechanisms. As mentioned in the response to Q2, besides layer-wise node features (embeddings), external factors may also induce diverse forms of relevance that describes the node-node relationship from various perspectives. Examples include the node-node correlations generated by node cluster embeddings (Eqn. (2)), and similarity in terms of input node features. It is known that previous graph attention networks compute attention coefficients in each layer solely based on node embeddings locating in the same layer of the GNN. As a result, how those diverse forms of node-node relevance that can be acquired outside of the GNN may affect the representation learning is not fully explored by the previous works. This motivates us to proposed conjoint attentions (CAs), which can pay attention to the similar embeddings of neighbors, as well as to the ones that share other forms of relevance with the central node. Based on the attention layer utilizing conjoint attention mechanisms, we construct Graph Conjoint Attention Networks (CATs) to learn node representations from those nodes that are heterogeneously relevant.
> >
> > As for the motivation of providing different conjoint attention mechanisms, we aim at allowing CATs to compute attentions coefficients which are dependent on the external-factor-caused relevance at different levels. Implicit direction (Eqn. (7)) provides a tender way to adjust node-embedding-based attentions (Eqn. (4)) using the attention scores that are acquired by external factors, while Explicit direction (Eqn. (8)) offers a radical method to compute attention coefficients based on node embeddings and external factors. The difference between these two conjoint attention mechanisms can be intuitively revealed by checking the extreme values. For example, if $s_{ij}$ is close to zero, the attention coefficient computed by Implicit direction is approaching to $f_{ij}$. In contrast, the attention score computed by Explicit direction is close to zero. If $s_{ij}$ is close to one, the attention coefficient computed by Implicit direction is a harmonic average between $s_{ij}$ and $f_{ij}$. But the attention score computed by Explicit direction is close to $f_{ij}$. It is seen that Explicit direction is more sensitive to $s_{ij}$, and how much $f_{ij}$ contributes to the conjoint attention coefficient is directly determined by $s_{ij}$. Thus, both higher $f_{ij}$ and $s_{ij}$, meaning that the relevance pertaining to node embeddings and external factors is homogeneous, can produce higher attention coefficients. Different from Explicit direction, Implicit direction aims at averaging the values of $f_{ij}$ and $s_{ij}$, Implicit direction may still pay some attention to the neighbor whose $s_{ij}$ is close to zero. Such differences make the two conjoint attentions mechanisms obtain different performances in our experiments.
> >
> > $Q5$. In section 3.1 the authors explain that conjoint attentions can capture “enlightenment outside of GNNs”. Again, it isn’t clear what that means. The authors use the word “enlightenment” to refer to the information that the proposed attention mechanism can use, but it isn’t clear what that means.
> >
> > $Response$: Thank you very much for the suggestion and we sincerely apologize for any confusions caused. Originally, we use the term “enlightenment” in the paper to provide a clear representation of the factors that can be acquired outside of the GNN that can then be used for computing the conjoint attentions. This might include various graph structural features in the papers mentioned by reviewer zBg1, and others, for example, the input textual features mentioned by Reviewer vEEu. Nonetheless, we noted that the term “enlightenment” may lead to possible misunderstanding on the core idea of our proposed work in the manuscript. Thus, following the reviewer’s suggestion, we will not introduce “enlightenment” as a new term in the revised manuscript. Instead, we will clearly illustrate what external factors are currently used by CATs to compute conjoint attentions utilizing commonly-used terms. They include node-node correlations generated by node cluster embeddings, node self-representation coefficients (Eqns. (2) and (3)), and similarities in terms of input node features. We will also introduce more methods together with examples to better clarify what factors outside of GNNs can be potentially used to compute conjoint attentions.
> >
> > $Q6$. In Equations (2) and (3) it seems that C_ij is capturing structural information from the adjacency matrix of the graph, but that isn’t explained.
> >
> > $Response$: Thank you for the suggestion. $\mathbf C_{ij}$ will be explained with more details in the final version of the paper to avoid any confusions. In the original version of the paper, we use two different methods, i.e., matrix factorizations (Eqn. (2)) [1] and self-expressiveness module (Eqn. (3)) [2], to learn $\mathbf C_{ij}$ from adjacency matrix. Given the properties of two learning methods, $\mathbf C_{ij}$ learned by Eqn. (2) represents the node-node correlation generated by node cluster embeddings. A higher value of $\mathbf C_{ij}$ learned by Eqn. (2) means a pair of nodes are very likely to belong to the same cluster. $\mathbf C_{ij}$ learned by Eqn. (3) represents the node self-representation coefficient. And a higher value of $\mathbf C_{ij}$ learned by Eqn. (3) means the global structure of node $i$ can be better represented by the global structure of node $j$, and consequently this pair of nodes are more structurally correlated. Besides, $\mathbf C_{ij}$ may also be obtained using other information, e.g., higher-order structural similarity between nodes, and the node-node similarity in terms of input node features. We have used CATs utilizing $\mathbf C$ computed based on the cosine similarity in terms of input node features to learn node representations in all the testing datasets. The corresponding results have been presented in the table above, and will also be included into the final version of the paper.
> >
> > $Q7$. In the experiments the authors evaluate the MF and SC variants of their method, introduced in Equations (2) and (3), but they don’t explain much about their motivation. What is the difference between them? They seem to get similar experimental results, why?
> >
> > $Response$: Thank you for the suggestion. Node cluster embeddings and node self-representation coefficients are established latent structures that can be learned from graph adjacency matrix. However, they are always overlooked or underexplored by previous attention-based GNN efforts. Aiming at paying more attention to those vertices that are correlated from the structural perspective, we treat the node-node correlations generated by node cluster embeddings and node self-representation coefficients as effective external factors and have CATs use them to compute the conjoint attentions.
> >
> > As mentioned in the paper, the proposed CATs make use of $\mathbf V$ to generate $\mathbf C$, i.e., $\mathbf C_{ij}$ = $\mathbf {VV}^T_{ij}$. Here $\mathbf V$ is an $N$-by-$C$ matrix, where $N$ and $C$ represent the numbers of nodes and classes in the graph, respectively. Given Eqn. (2) (MF), we know that $\mathbf C_{ij}$ is used to approximate $\mathbf A_{ij}$. Thus, $\mathbf C_{ij}$ generated by node cluster embeddings ($\mathbf V$) mainly consider the local structure around the node. Different from MF, in the self-expressiveness module (Eqn. (3)), $\mathbf A_{i, :}$ is approximated by $\sum_{j \ne i} \mathbf C_{ij} \mathbf A_{j, :}$,:, where $\mathbf A_{i, :}$ represents the $i$th row of $\mathbf A$. Thus, $\mathbf C_{ij}$ in Eqn. (3) (node self-representation coefficients) represents how the global structure of $i$th node can be represented by the global structure of node $j$. Although both node-node correlations regarding to node cluster embeddings and node self-representation coefficients are both learned from adjacency matrix, the considerations of local and global structure to generate $\mathbf C$ potentially lead the slightly different variants and hence performances of CATs.

---

> > > ### Author Response · Authors · 2021-08-09
> > > **Response to the comments from Reviewer dyaY - Part 3**
> > >
> > > $Q8$. In Section 5.4 the authors say that the “external enlightenment C” is learnable, and that it can be simultaneously learned during the training of CATs. But why is it learnable? Does it have any learning parameters? Why can’t it be computed before training?
> > >
> > > $Response$: Thank you for the suggestion. The point raised will be addressed in the final version of paper. Specifically, let us take Eqn. (2) as an example to show why $\mathbf C$ is learnable. As defined in the manuscript, $\mathbf C$ represents the node-node correlations obtained outside of the GNN. To reduce the computational burden, we additionally define an $N$-by-$C$ matrix $\mathbf V$ to generate $\mathbf C$, where $N$ and $C$ represent the numbers of nodes and classes in the graph respectively. Thus, $\mathbf C_{ij}$ = $\mathbf {VV}^T_{ij}$, and $\mathbf C_{ij}$ can be learned by minimizing the difference between $\mathbf A_{ij}$ and $\mathbf {VV}^T_{ij}$. As $\mathbf V$ is defined, CATs need additional learning parameters while training the neural network. In Table 3 of the appendix, we compare the number of parameters between GAT and CAT. As shown in the table, CAT requires more learning parameters to complete the training task, when both GAT and CAT have the same neural network structure. It should be noted that $\mathbf C$ can be obtained prior to the training of CATs. This is especially meaningful where computational power is limited. We combine the learning of $\mathbf C$ with the training of CATs to make the proposed framework more “integrated.”
> > >
> > > $Q9$. In Figure 3, it appears that the value of λ doesn’t have an effect, why is that the case? It isn’t explained. What should be the behaviour C_ij of as λ increases (or decreases)?
> > >
> > > $Response$: Obtaining the results depicted in Fig. 3 might be due to the following reasons. As mentioned in Section 5.4 of the manuscript, the loss function of the proposed CATs can be $L = L_{predict} + \lambda L_{ext}$, where $L_{predict}$ and $L_{ext}$ respectively represent the task-specific loss of the GNN and the loss for learning external factors (i.e., $\mathbf C$) for computing conjoint attentions. In this paper, we mostly use semi-supervised learning tasks to test the effectiveness of different approaches. This means only a very low proportion of node labels (e.g., 140 out of 2708 in dataset Cora) are used to compute $L_{predict}$ and consequently only the gradients related to this small number of labeled nodes will be computed in the back propagation stage of CATs in each training epoch. Besides, a proportion of the gradients related to the labeled nodes might still be set to zero due to the dropout mechanism in each layer of the GNN. Given the mentioned property of learning tasks and the dropout mechanism used by GNN, in each training epoch, the gradients related to $L_{predict}$ contribute very limitedly to the learning of $\mathbf C_{ij}$. In contrast, $L_{ext}$ sums all the $\mathbf C_{ij}$s and we do not apply dropout on the training of $L_{ext}$, i.e., the learning of $\mathbf C_{ij}$. As a result, the learning of $\mathbf C_{ij}$ is dominantly determined by $L_{ext}$, i.e., Eqns. (2) or (3) in the manuscript, and the proposed CATs are less sensitive to the changes of $\lambda$. Changing the settings of $\lambda$ may only affect the convergence of learning $\mathbf C_{ij}$, i.e., the gradients for learning $\mathbf C_{ij}$ resulted from $L_{ext}$ in each epoch, can be amplified (shrunk) as $\lambda$ increases (decreases).  We will include the above discussions into the final version of the paper.
> > >
> > > $Q10$. What are the different attention factors compared in Figure 2? What is the difference between SC and I-SC?
> > >
> > > $Response$: Fig. 2 compares the performances of CATs utilizing different factors to compute attention scores. SC means CATs only use $\mathbf C$ learned by Eq. (3) to compute attention score. While I-SC means CATs make use of the Implicit direction (Eqn. (7) in the manuscript) to compute attention scores. And $\mathbf C$ used by I-SC is learned via Eqn. (3). As the figure shows, CATs using conjoint attentions (I-SC, E-SC, I-MF, E-MF) may outperform CATs using only $\mathbf C$ or node embeddings (F, equivalent to GAT) to compute attention scores. The meaning of those abbreviations mentioned by the reviewer will be clearly explained in the final version of the paper.
> > >
> > > $Q11$. To summarize, the method seems to perform well, and the authors do a good evaluation in different datasets, but it is not clear why this method should work better than GATs. The authors keep mentioning "enlightenment", a concept that is not explained. The clarity of the paper and the quality of the explanations should improve in order to be accepted.
> > >
> > > $Response$: Thanks very much for raising the suggestions to improve the paper. We will try our best to improve the clarity of the revised manuscript by providing more explanations with examples to the key concepts used.
> > >
> > > The term “enlightenment” was originally introduced in this paper to provide a clearer representation of the external factors that can be acquired outside of the GNN, and which can then be used for the computation of conjoint attentions. This might include various graph structural features, and others, for example, the input textual features. Nonetheless, we noted that the term “enlightenment” may lead to possible misunderstanding on the core idea of our proposed work in the manuscript. Thus, following the reviewer’s suggestion, we will not introduce “enlightenment” as a new term in the final version of the paper. Instead, external factors that are used by CATs to compute conjoint attentions will be clearly introduced by utilizing commonly used terms. Such external factors include node-node correlations generated by the cluster embeddings (Eqn. (2)), node self-representation coefficients (Eqn. (3)), and similarities regarding to input node features. More external factors that can be potentially used to compute conjoint attentions will also be introduced with examples in the final version of the paper.
> > >
> > > Although many other external factors can be used by CATs, here let us take the node-node correlations pertaining to node cluster embeddings (Eqn. (2)) as an example to explain why the proposed CATs perform better than GAT, simultaneously avoiding any potential confusions. It is known that clusters are meaningful latent structures hidden in the graph and those nodes in the same cluster potentially share higher cohesiveness. If the cluster structure is considered, an attention-based GNN is able to learn node representations that tends towards using embeddings of those neighbors in the same cluster. In practice, such node representations might be learned from those articles whose topics are correlated and that are frequently cited by other similar articles in citation networks, or from those social network users who frequently interact with each other in the same social group as well as sharing correlated hobbies in social graphs. Though such external factors like cluster structures might be beneficial to representation learning, previous graph attention networks cannot appropriately use them together with node embeddings to compute the attention coefficients. We therefore propose conjoint attention (CA) mechanisms and utilize them to construct Graph Conjoint Attention Networks (CATs). Paying attention to the similar embeddings of neighbors, the proposed conjoint attentions also can pay attention to those neighbors sharing other types of relevance with the central node, when computing attention coefficients. Meaningful node representations are expected to learn from those nodes that are heterogeneously relevant. In Section 4 of the manuscript, we prove that the expressive power of the proposed conjoint attentions is stronger than empirical graph attentions. And the experimental results also demonstrate and validate that the proposed CATs are performing better than the representative attention-based GNN, i.e., GAT. We additionally present the results obtained by CATs using node-node similarity regarding to input features, which has shown to be another effective external factor to compute conjoint attentions. These results would further demonstrate and validate the increase performance efficacies of the proposed CATs.

---

> > > > ### Author Response · Authors · 2021-08-11
> > > > **Response to the comments from Reviewer dyaY - Part 4**
> > > >
> > > > $Q12$. Why the different methods for computing C_ij are chosen?
> > > >
> > > > $Response$: Thank you very much for pointing this out. The reasons that different methods for computing $\mathbf C_{ij}$ are used will be explained in the final version of the paper. In the proposed CATs, $\mathbf C_{ij}$s represent external factors that can be acquired outside of the GNN and can then be used to compute conjoint attentions. It is known that previous attention-based GNNs compute attention coefficients solely based on the relevance of layer-wise node embeddings (features). While external factors that may represent diverse forms of relevance between pairwise nodes are overlooked or underexplored. We therefore propose conjoint attention mechanisms to make use of both node embeddings in the GNN and external factors to compute attention coefficients. There are many types of well-established external factors that can represent diverse forms of node-node relevance, other than that caused by the node embeddings in the GNN. Examples of such external factors include node-node correlations generated by node cluster embeddings (Eqn. (2)), node self-representation coefficients (Eqn. (3)), and node-node similarity in terms of input node features. To investigate whether the proposed conjoint attention mechanisms are effective when various external factors are considered, we select different methods to acquire different external factors (i.e., $\mathbf C_{ij}$s) and have CATs use them to compute conjoint attention coefficients for node representation learning. The corresponding experimental results have demonstrated that CATs may perform robustly when using different external factors that are either structure related (node-node correlations generated by node cluster embeddings and node self-representation coefficients), or node feature related (node-node similarity in terms of input node features, see the results in the Table above) to compute conjoint attention coefficients for node representation learning.
> > > >
> > > > [1] J. Yang, and J. Leskovec, Overlapping community detection at scale: a nonnegative matrix factorization approach. WSDM 2013, 587-596, 2013.
> > > >
> > > > [2] E. Elhamifar and R. Vidal, Sparse subspace clustering: Algorithm, theory, and applications. IEEE transactions on pattern analysis and machine intelligence, 35(11):2765–2781, 2013.

---

> > > > > ### Comment · Reviewer_dyaY · 2021-08-22
> > > > > **Good rebuttal**
> > > > >
> > > > > Thank you for your detailed response, it answers my questions, so I will raise my score and vote for acceptance, assuming that the authors will fix the clarity issues of some of the explanations in the paper.

---

> > > > > > ### Author Response · Authors · 2021-08-23
> > > > > > **Sincere gratitude for your careful review**
> > > > > >
> > > > > > Thank you very much again for the detailed review and constructive comments. All the clarity issues that have been pointed out by the reviewers will be addressed in the revised version of the paper.

---

### Official Review · Reviewer_zBg1 · 2021-07-16

**Rating:** 6
**Confidence:** 4

**Summary:**

The authors propose Graph ConJoint Attention Networks (CATs) as a mechanism for explicitly incorporating structural information in the GATs attention mechanism.

CATs separately learn an "enlightenment" attention matrix which exploits structural correlations between the vertices (based on the adjacency matrix).

Such a matrix is then recombined with the attention matrix computed by vanilla GATs, yielding updated attentional coefficients which empirically improve on GATs on several benchmark datasets, and can be proven (with small modifications) to satisfy the equivalence with the 1-WL test, which also answers a somewhat-open question on GATs' expressive power.

**Limitations And Societal Impact:**

No concerns here.

**Main Review:**

Overall, I found the work's theoretical contribution to be impressive, and the comparisons to baseline approaches to be relevant.

I believe that the work is worthy of acceptance, but I would like to see a few more experiments, comparisons and clarity improvements before giving a higher score.

Specifically, the proposed structural attention is one way of injecting structural information into the attentive mechanism, but is not the only one. Many prior works, that e.g. take the top/bottom-k eigenvectors of the Graph Laplacian, or a structural embedding vector such as DeepWalk/node2vec/struct2vec/LINE, could be meaningful baselines. This is especially given that recently a deep connection between such works and matrix factorisation has been uncovered (see NetMF; Qiu et al., 2017).

Could the authors compute these additional features, and include a GAT baseline that takes them in concatenated alongside the usual node features? This will strengthen the claim about the impact of the proposed CAT mechanism.

I found the paper at times very unclear and difficult to follow. The second sentence in the abstract for example:

"Different from previous graph attentions computing weights for feature aggregation solely bestowing layer-wise node features propagated inside GNN, CAs are capable of adaptively leveraging heterogeneous learnable factors either inside or outside of GNNs to compute attention scores."

It is very unclear what the authors want to say in this sentence, and in either case it would be better splitting it into multiple.

Lastly, I disagree with using the term "enlightenment" to refer to graph-structural features. They are very commonly used in many other established papers (e.g. see the ones I referred to above) and there is no need to overload a new term to illustrate the same (or very similar) concept.

I invite the authors to carefully rewrite some parts of the paper in light of the comment above, to make it easier to follow.

==== UPDATE Post-rebuttal:
The authors have carefully responded to my comments and included new experiments that further elucidate the gains of their method. With faith that they will rewrite their paper to be more clear, and avoid overloading existing terms, I will upgrade my score to a 6.

**Time Spent Reviewing:**

2

---

> ### Author Response · Authors · 2021-08-09
> **Response to the comments from Reviewer zBg1**
>
> $Q1$. Specifically, the proposed structural attention is one way of injecting structural information into the attentive mechanism, but is not the only one. Many prior works, that e.g. take the top/bottom-k eigenvectors of the Graph Laplacian, or a structural embedding vector such as DeepWalk/node2vec/struct2vec/LINE, could be meaningful baselines. This is especially given that recently a deep connection between such works and matrix factorisation has been uncovered (see NetMF; Qiu et al., 2017). Could the authors compute these additional features, and include a GAT baseline that takes them in concatenated alongside the usual node features? This will strengthen the claim about the impact of the proposed CAT mechanism.
>
> $Response$: As highlighted by the reviewer, directly concatenating input node features and structural embeddings is indeed an effective way to make current GNNs more structural bias. We will discuss such methods for injecting structural information into GNNs in our final version of the paper by referencing and citing all the articles mentioned by the reviewer. Following the suggestion of the reviewer, we select three representative methods including $k$-eigenvectors of graph Laplacian, Deepwalk, and NetMF to learn structural node embeddings in all the datasets. All the methods for learning structural node embeddings are configured using the recommended settings and the output dimensionality of the structural node embeddings is configured as equal to the number of classes of each testing dataset. We then have GAT performs different learning tasks using the concatenation of original input features and structural embeddings. The performance comparison between GAT using different structural embeddings and CATs is summarized in the table below. As the Table shows, the learning performance is improved in most datasets when GAT uses the concatenation of original node features and various structural embeddings. However, in most datasets, such performance improvement is not as significant as that obtained by CATs. Different from directly concatenating the original node input features and structural embeddings for learning node representations, CATs provide a smooth means to compute attention coefficients that would jointly consider the diverse relevance regarding to both layer-wise node embeddings and external factors, like the correlations generated by the node cluster embeddings considered in this paper. As a result, CATs can learn node representations from those nodes that are heterogeneously relevant and attain notable learning performance on all the testing datasets. The corresponding results will be included into the final version of the paper.
>
> |--------------------|-----------------|-----------------|-----------------|-----------------|-----------------------|
>
> | Node Classification|   Cora          |   Cite          |   Pubmed        |   CoauthorCS    |   OGB-Arxiv           |
>
> |--------------------|-----------------|-----------------|-----------------|-----------------|-----------------------|
>
> |   GAT              |   83.84 (0.61)  |   70.36 (0.42)  |   81.50 (0.47)  |   92.80 (0.41)  |   72.39 (0.07)  |
>
> |   GAT-k-Lap        |   84.1 (0.24)   |   71.18 (0.52)  |   82.56 (0.30)  |   92.70 (0.31)  |   72.47 (0.06)  |
>
> |   GAT-NetMF        |   84.44 (0.19)  |   70.94 (0.16)  |   81.90 (0.33)  |   93.16 (0.27)  |   72.42 (0.08)  |
>
> |   GAT-Deep         |   83.68 (0.67)  |   69.70 (0.57)  |   80.13 (0.26)  |   92.93 (0.17)  |   72.79 (0.09)  |
>
> |   CAT-I-MF         |   85.38 (0.16)  |   73.22 (0.19)  |   83.90 (0.24)  |   93.74 (0.14)  |   72.89 (0.06)  |
>
> |   CAT-I-SC         |   85.50 (0.22)  |   73.18 (0.22)  |   84.28 (0.20)  |   93.70 (0.11)  |   72.85 (0.04)  |
>
> |   CAT-E-MF         |   85.56 (0.19)  |   73.24 (0.21)  |   83.60 (0.17)  |   93.40 (0.12)  |   72.81 (0.09)  |
>
> |   CAT-E-SC         |   85.40 (0.36)  |   73.02 (0.24)  |   84.02 (0.24)  |   93.30 (0.11)  |   72.83 (0.11)  |
>
> |--------------------|-----------------|-----------------|-----------------|-----------------|-----------------------|
>
> |   Node Clustering  |   Cora          |   Cite          |   Pubmed        |   CoauthorCS    |   OGB-Arxiv           |
>
> |--------------------|-----------------|-----------------|-----------------|-----------------|-----------------------|
>
> |   GAT              |   81.39 (0.18)  |   69.20 (0.28)  |   80.88 (0.33)  |   90.09 (0.15)  |   76.04 (0.38)  |
>
> |   GAT-k-Lap        |   80.66 (0.31)  |   69.56 (0.34)  |   81.59 (0.09)  |   89.83 (0.18)  |   76.21 (0.06)  |
>
> |   GAT-NetMF        |   81.75 (0.26)  |   68.96 (0.21)  |   81.74 (0.18)  |   89.85 (0.21)  |   76.06 (0.07)  |
>
> |   GAT-Deep         |   81.08 (0.41)  |   68.27 (0.06)  |   80.55 (0.11)  |   89.70 (0.27)  |   76.91 (0.15)  |
>
> |   CAT-I-MF         |   82.17 (0.11)  |   71.15 (0.12)  |   82.77 (0.07)  |   90.26 (0.22)  |   77.72 (0.07)  |
>
> |   CAT-I-SC         |   82.26 (0.13)  |   71.17 (0.15)  |   82.86 (0.07)  |   90.29 (0.21)  |   77.01 (0.16)  |
>
> |   CAT-E-MF         |   81.98 (0.19)  |   71.21 (0.12)  |   82.40 (0.08)  |   89.66 (0.22)  |   76.93 (0.16)  |
>
> |   CAT-E-SC         |   82.01 (0.24)  |   71.11 (0.24)  |   82.61 (0.14)  |   89.72 (0.15)  |   76.98 (0.08)  |
>
> |--------------------|-----------------|-----------------|-----------------|-----------------|-----------------------|
>
> $Q2$. I found the paper at times very unclear and difficult to follow. The second sentence in the abstract for example:
> "Different from previous graph attentions computing weights for feature aggregation solely bestowing layer-wise node features propagated inside GNN, CAs are capable of adaptively leveraging heterogeneous learnable factors either inside or outside of GNNs to compute attention scores." It is very unclear what the authors want to say in this sentence, and in either case it would be better splitting it into multiple.
>
> $Response$: Thank you very much for pointing this out. The contents highlighted represented our first attempt to illustrate the core difference between previous graph attentions and our proposed conjoint attentions. We realized that this has not being clear enough and we apologize for the confusion caused. Thus, we would like to better clarify it is known that previous methods compute the coefficients solely based on the node embeddings (features) in each layer of the GNN. As these node embeddings are learned within the GNN structure, we regard them as internal factors for computing attention coefficients. In contrast, our proposed conjoint attentions can leverage not only node embeddings in the GNN, but also the factors learned outside of the GNN, e.g., node-node correlations caused by node cluster embeddings learned from adjacency matrix (Eqn. (2)) to compute the attention scores. We regard those factors that can be acquired outside of the GNN as external factors. As requested by the reviewer, the mentioned content will be rewritten and better clarified to make the paper easier to follow and comprehend. Besides, to avoid any potential confusions, in the revised manuscript, we will clearly define what inter and external factors stand for and will directly illustrate what factors, e.g., node embeddings, correlations generated by node cluster embeddings (Eqn. (2)), and node self-representation coefficients (Eqn. (3)) [1] are used by the proposed conjoint attention mechanisms whenever it needs.
>
> $Q3$. Lastly, I disagree with using the term "enlightenment" to refer to graph-structural features. They are very commonly used in many other established papers (e.g. see the ones I referred to above) and there is no need to overload a new term to illustrate the same (or very similar) concept.
>
> $Response$: Thank you for the suggestion. The term “enlightenment” was originally introduced in this manuscript to provide a clearer representation of the external factors that can be learned or obtained outside of the GNN which can then be used for the computation of conjoint attentions. This might include various structural embeddings in the papers mentioned by the reviewer, and others, for example, the input textual features mentioned by Reviewer vEEu. Nonetheless, we noted that the term “enlightenment” may lead to possible misunderstanding on the core idea of our proposed work in the manuscript. Thus, following the suggestion of the reviewer, we will not introduce “enlightenment” as a new term in the revised manuscript. Instead, we will use more commonly-used terms to illustrate what is currently used by CATs to compute conjoint attentions, i.e., node-node correlations generated by node cluster embeddings (Eqn. (2)) [1], self-representation coefficients (Eqn. (3)) [2], and similarities in terms of input node features. We will also introduce more methods together with examples to better clarify and illustrate what are the factors outside of GNNs that can be used to compute conjoint attentions.
>
> $Q4.$ I invite the authors to carefully rewrite some parts of the paper in light of the comment above, to make it easier to follow.
>
> $Response$: Thank you very much for your constructive suggestions to improve the paper. We will revise and improve the readability of the paper in all the parts that have been highlighted and also further proofread and improve all sections of the paper.
>
> [1] J. Yang, and J. Leskovec, Overlapping community detection at scale: a nonnegative matrix factorization approach. WSDM 2013, 587-596, 2013.
>
> [2] E. Elhamifar and R. Vidal, Sparse subspace clustering: Algorithm, theory, and applications. IEEE transactions on pattern analysis and machine intelligence, 35(11):2765–2781, 2013.

---

> > ### Comment · Reviewer_zBg1 · 2021-08-09
> > **Score upgrade**
> >
> > Thank you for carefully responding to all of my comments.
> > While the gains of CAT appear more modest in comparison when compared against GATs with structural features, a gain seems to be there, and the added experiments are certainly useful.
> >
> > With faith that you will carefully correct the writeup for clarity and avoid overloading terms that don't need to be overloaded, I will raise my score to a weak accept. Good luck!

---

> > > ### Author Response · Authors · 2021-08-09
> > > **Sincere gratitude**
> > >
> > > We would like to express our sincere gratitude to you and all the other reviewers for the careful review and constructive comments. We will revise our paper following the comments made by all the reviewers.

---

### Official Review · Reviewer_vEEu · 2021-07-17

**Rating:** 7
**Confidence:** 4

**Summary:**

This paper aims to establish a new type of attention mechanism for GNNs. Authors propose an attention mechanism which can utilize structural interactions between nodes instead of focusing only on given node features. Given the new attention authors use it create Conjoint Graph Attention Networks to solve node classification and node clustering tasks.

Authors first calculate an external ( w.r.t GNNs ) information matrix and propose two forms in this paper based on Matrix Factorization and Coefficient of Self-Expresiveness. The *Conjoint Attention* layer then calculates two types of attention, using external information matrix and using the node features ( any attention mechanism can be used here, authors use attention mechanism similar to GAT ). Two strategies are used to combine the different attentions calculated in previous step, Implicit and Explicit. Weighted aggregation is used to get the final node embeddings, with an addition of a correction factor.

Experimentation is conducted of several publicly available datasets, and comparisons to several baselines are included which gives a good idea of the expressive power of the proposed attention mechanism.

**Limitations And Societal Impact:**

Questions and Comments
- In Section 3.2, please define the matrix V and how is it defined.
- It is stated in the paper that external information matrix C can be calculated based on any interaction between nodes which is useful in node classification tasks which have some textual data available for each nodes. In such cases, I would assume that using conjoint attention principle to calculate an attention score between texts of two nodes will be useful in the final task. Would you consider including such an experiment in the paper?


**Main Review:**

- Authors propose to use the structural graph properties in addition to node features, to calculate separate attention scores. The motivation behind this type of attention is well established.
- Thorough experimental analysis is done to establish the advantage of using the proposed attention mechanism.
- The writing of the paper is okay but can be improved by giving examples at several places. For instance, authors talk about " heterogeneous learnable factors either inside or outside GNNs " starting from Abstract of the paper but it is only introduced in the method section. Please provide simple examples in order to make the method more approachable.


**Time Spent Reviewing:**

5

---

> ### Author Response · Authors · 2021-08-09
> **Response to the comments from Reviewer vEEu**
>
> $Q1$. The writing of the paper is okay but can be improved by giving examples at several places. For instance, authors talk about " heterogeneous learnable factors either inside or outside GNNs " starting from Abstract of the paper but it is only introduced in the method section. Please provide simple examples in order to make the method more approachable.
>
> $Response$: Thank you for the suggestion. As requested by the reviewer, more examples of heterogeneous learnable factors either inside or outside GNNs will be provided in both the abstract and introduction to bring better understandings of the proposed method. Specifically, in this manuscript, learnable factors inside GNNs (internal factors) that can be used to compute attention coefficients are the low-dimensional node embeddings (also known as layer-wise node features) learned in each GNN layer. While, external factors represent the parameters that can be learned or obtained outside of empirical graph attention networks (GATs) and can then be used for the computation of conjoint attentions. Examples include node-node correlations in terms of cluster embeddings learned from adjacency matrix using matrix factorizations (Eqn. (2)) [1], node self-representation coefficients learned from adjacency matrix using self-expressiveness module (Eqn. (3)) [2], higher order correlations pertaining to graph connectivity, vertex-vertex similarity in terms of input node features, vertex-vertex fused correlations regarding to both graph connectivity and node features, node-node feature diffusion, and others. We believe these mentioned factors can provide a different view on measuring the relative significance (i.e., attentions) between pairwise nodes. As a result, the proposed CATs can pay attention to those nodes that share relevant node embeddings with the center node, and are related to the center node with respect to the external factors.
>
> Although many other external factors can be used by CATs, here the node-node correlations generated by node cluster embeddings (Eqn. (2)) is used as an example for illustration purpose to avoid any potential confusions while bringing about better understandings of the proposed approach. Clusters are well-recognized latent structures hidden in the graph and they can be learned from adjacency matrix. Nodes in the same cluster share more group cohesiveness, which is frequently observed in many real-world scenarios, for example, scientific articles are often cited by others that investigate the related topics, and social network users that share similar interests lean towards joining the same online groups. Given Eqn. (2), a higher value of $\mathbf C_{ij}$ means a pair of nodes are more likely to belong to the same group, and therefore they are structurally correlated. However, such node cluster embeddings that can be learned from adjacency matrix, rather than GNN, cannot be appropriately used by empirical attention-based GNNs to compute attention coefficients. We therefore propose conjoint attentions, allowing CATs to learn node representations from those nodes that are relevant with respect to both node embeddings within CATs (internal factors) and cluster embeddings (external factors) learned outside of the CATs. As a result, the performance of CATs is shown to be better than previously reported attention-based GNNs.
>
> [1] J. Yang, and J. Leskovec, Overlapping community detection at scale: a nonnegative matrix factorization approach. WSDM 2013, 587-596, 2013.
>
> [2] E. Elhamifar and R. Vidal, Sparse subspace clustering: Algorithm, theory, and applications. IEEE transactions on pattern analysis and machine intelligence, 35(11):2765–2781, 2013.
>
> $Q2$. In Section 3.2, please define the matrix V and how is it defined.
>
> $Response$: In this paper, $\mathbf V$ is an $N$-by-$C$ matrix, where $N$ and $C$ represent the numbers of nodes and classes of the graph, respectively. The proposed CATs make use of $\mathbf V$ to generate $\mathbf C$, i.e., $\mathbf C_{ij}$ = $\mathbf {VV}^T_{ij}$. Referring to Eqns. (2) and (3) the generated $\mathbf C$ captures the correlations pertaining to node cluster embeddings and self-representation coefficients, respectively, that are learned from the adjacency matrix. As suggested by the reviewer, the detailed definition of matrix $\mathbf V$ will be introduced in the final version of the paper.
>
> $Q3$. It is stated in the paper that external information matrix C can be calculated based on any interaction between nodes which is useful in node classification tasks which have some textual data available for each nodes. In such cases, I would assume that using conjoint attention principle to calculate an attention score between texts of two nodes will be useful in the final task. Would you consider including such an experiment in the paper?
>
> $Response$: As input node features can also be seen as factors outside of the GNN, CATs can utilize them to compute the conjoint attentions. We have obtained $\mathbf C$ based on cosine similarity in terms of input node features and then CATs can use the obtained $\mathbf C$ and layer-wise node embeddings to compute the conjoint attentions. The performance comparisons between CATs using similarity of input node features, and GAT are briefly summarized below. As can be observed, CATs outperform GAT in most of the datasets when performing node classification and clustering tasks.
>
> |-------------------------|-------------------|------------------|------------------|------------------|------------------|
>
> |     Node Classification |          Cora     |         Cite     |        Pubmed    |      CoauthorCS  |       OGB-Arxiv  |
>
> |-------------------------|-------------------|------------------|------------------|------------------|------------------|
>
> |             GAT         |      83.84 (0.61) |     70.36 (0.42) |     81.50 (0.47) |     92.80 (0.41) |     72.39 (0.07) |
>
> |           CAT-I-FS      |      84.86 (0.30) |      72.54 0.47) |     82.79 (0.34) |     93.08 (0.24) |     72.57 (0.02) |
>
> |           CAT-E-FS      |     84.15 (0.23)  |     71.68 (0.45) |     82.33 (0.27) |     93.33 (0.23) |     72.52 (0.03) |
>
> |-------------------------|-------------------|------------------|------------------|------------------|------------------|
>
> |       Node Clustering   |          Cora     |         Cite     |        Pubmed    |      CoauthorCS  |       OGB-Arxiv  |
>
> |-------------------------|-------------------|------------------|------------------|------------------|------------------|
>
> |             GAT         |      81.39 (0.18) |     69.20 (0.28) |     80.88 (0.33) |     90.09 (0.15) |     76.04 (0.38) |
>
> |           CAT-I-FS      |      81.83 (0.23) |     70.36 (0.43) |     81.89 (0.37) |     90.20 (0.17) |     76.33 (0.03) |
>
> |           CAT-E-FS      |      81.50 (0.26) |     70.37 (0.43) |     81.91 (0.34) |     90.01 (0.19) |     76.18 (0.33) |
>
> |-------------------------|-------------------|------------------|------------------|------------------|------------------|

---

> > ### Comment · Reviewer_vEEu · 2021-08-19
> > **Revised Score**
> >
> > Thank you for responding to all the comments in detail. I am updating my score to a 7 and hope that authors will make the aforementioned changes in the final version. Good Luck!

---

> > > ### Author Response · Authors · 2021-08-20
> > > **Sincere gratitude**
> > >
> > > Thank you very much again for the careful review and constructive comments. All the aforementioned changes will be made in the revised paper.

---

### Author Response · Authors · 2021-08-10
**Summary of the response to all the reviewers' comments**

We would like to express our sincere gratitude to all the reviewers for their careful review and constructive comments. Here we briefly summarize our response to all the comments from all the reviewers.

1.	All the confusing contents that are pointed out by the reviewers have been explained by providing more details and examples.
2.	The motivation of the proposed approach and the reasons that the proposed method performs better than other related works are explained in detail.
3.	The core differences between the proposed method and other related works have been illustrated with more details and examples.
4.	As requested, some modules of the proposed approach are clearly defined.
5.	Further discussions on the limitations of the proposed method and the future works are performed.
6.	As suggested, the potential reasons that lead the presented experimental results are given.
7.	All the suggested experiments have been performed and the corresponding results have been reported.
8.	How we will improve the paper according to all the reviewers’ suggestions has been illustrated.

We thank all the reviewers again and we will carefully revise the paper following all the suggestions raised.

---

### Decision · Program_Chairs · 2021-09-27

**Decision:**

Accept (Poster)

**Comment:**

There is general consensus among the reviewers that the paper should be accepted.

The authors made extensive effort to answer reviewer comments and provide additional empirical results.

We expect the authors to implements all clarity improvements, as requested by the reviewers.

Figure 1 looks pixelized, it should be converted to vector graphics.